# Shifting food web structure during dam removal—Disturbance and recovery during a major restoration action

**Sarah A. Morley**[1]*, **Melissa M. Foley**[2¤], **Jeffrey J. Duda**[3], **Mathew M. Beirne**[4], **Rebecca L. Paradis**[4], **Rachelle C. Johnson**[3], **Michael L. McHenry**[4], **Mel Elofson**[4], **Earnest M. Sampson**[4], **Randall E. McCoy**[4], **Justin Stapleton**[4], **George R. Pess**[1]

**1** Fish Ecology Division, Northwest Fisheries Science Center, National Marine Fisheries Service, National Oceanic and Atmospheric Administration, Seattle, Washington, United States of America, **2** United States Geological Survey, Pacific Coastal and Marine Science Center, Santa Cruz, California, United States of America, **3** United States Geological Survey, Western Fisheries Research Center, Seattle, Washington, United States of America, **4** Natural Resources Department, Lower Elwha Klallam Tribe, Port Angeles, Washington, United States of America

¤ Current address: San Francisco Estuary Institute, Richmond, California, United States of America
* Sarah.Morley@noaa.gov

**Data Availability Statement:** Data generated in this study and associated efforts monitoring environmental metrics before and during Elwha dam removal are available in: Foley MM, Shafroth PB, Beirne MM, Paradis RL, Ritchie AC, and Duda

## Abstract

We measured food availability and diet composition of juvenile salmonids over multiple years and seasons before and during the world's largest dam removal on the Elwha River, Washington State. We conducted these measurements over three sediment-impacted sections (the estuary and two sections of the river downstream of each dam) and compared these to data collected from mainstem tributaries not directly affected by the massive amount of sediment released from the reservoirs. We found that sediment impacts from dam removal significantly reduced invertebrate prey availability, but juvenile salmon adjusted their foraging so that the amount of energy in diets was similar before and during dam removal. This general pattern was seen in both river and estuary habitats, although the mechanisms driving the change and the response differed between habitats. In the estuary, the dietary shifts were related to changes in invertebrate assemblages following a hydrological transition from brackish to freshwater caused by sediment deposition at the river's mouth. The loss of brackish invertebrate species caused fish to increase piscivory and rely on new prey sources such as plankton. In the river, energy provided to fish by Ephemeroptera, Plecoptera, and Trichoptera taxa before dam removal was replaced first by terrestrial invertebrates, and then by sediment-tolerant taxa such as Chironomidae. The results of our study are consistent with many others that have shown sharp declines in invertebrate density during dam removal. Our study further shows how those changes can move through the food web and affect fish diet composition, selectivity, and energy availability. As we move further along the dam removal response trajectory, we hypothesize that food web complexity will continue to increase as annual sediment load now approaches natural background levels, anadromous fish have recolonized the majority of the watershed between and above the former dams, and revegetation and microhabitats continue to develop in the estuary.

JJ. Ecological parameters in the Elwha River estuary before and during dam removal (ver. 2.0, August 2020): U.S. Geological Survey data release. 2020. Available: https://doi.org/10.5066/F75B00N4 Morley, SA, Duda JJ, Johnson RC, McHenry ML, Elofson M, Sampson EM, and Pess GR. 2020. Fish Diet and Invertebrate Drift Data in the Elwha River Watershed Before and During Dam Removal. U.S. Department of Commerce, NOAA Data Report NMFS-NWFSC-DR-2020-02. https://doi.org/10.25923/eg1s-t677.

**Funding:** Funding for this work was supported by grants to: MMB, RLP, and MLM by the U.S. Environmental Protection Agency (EPA, https://www.epa.gov/): Wetlands Assessment Grant (CFDA#66.461), Puget Sound Partnership Puget Sound Protection and Restoration Tribal Assistance Program (CFDA#66.121), Technical Investigations and Implementation Assistance Program (CFDA#66.123), and Indian Environmental General Assistance Program (CFDA #66.926). MMB by the U.S. Geological Survey's (USGS, https://www.usgs.gov/): Coastal and Marine Geology and Environments programs, and the USGS Mendenhall Postdoctoral Research Fellowship Program. JJD by USGS (https://www.usgs.gov/) Coastal Habitats in Puget Sound Program. GRP by National Oceanic and Atmospheric Administration (NOAA, https://www.noaa.gov/) Open Rivers Initiative (CFDA#11.463), NOAA Restoration Center, Office of Habitat Conservation. The funders had no role in study design, data collection and analysis, decision to publish, or preparation of the manuscript.

**Competing interests:** The authors have declared that no competing interests exist.

## Introduction

Over 1500 dams have been removed in the United States to date, with the rate of removal doubling every decade since the mid-1960s [1,2]. A central goal of dam removal is typically to restore natural ecosystem processes, but short-term negative ecological impacts can occur when high volumes of impounded sediment are released downstream during removal [3]. Such was the case with the Elwha River dam removals in Washington State, among the largest managed sediment releases in history [4,5].

Two hydroelectric dams were built on the Elwha River in the early 1900s to provide power to the north Olympic Peninsula (Fig 1). The first dam completed in 1913 was the 32-m high Elwha Dam at river kilometer (rkm) 7.9, forming the Lake Aldwell reservoir (10 million m$^3$ capacity). In 1927, construction was completed on the 64-m tall Glines Canyon Dam at rkm 21.6, forming Lake Mills (50 million m$^3$). For nearly a century, these dams blocked upstream migration of anadromous fish and severely disrupted downstream transport of sediment and habitat-forming large woody debris [6,7].

At the time of dam removal, a combined 30 million tonnes (Mt) of sediment had accumulated behind the two Elwha River dams (7 Mt and 23 Mt in Lake Aldwell and Lake Mills respectively; [8]). Downstream of each dam, the river bed coarsened and coastal beaches eroded [9,10]. These physical changes further affected fish populations [7,11], as did associated changes to benthic invertebrates and primary producers [12].

In 1992, the Elwha River Ecosystem and Fisheries Restoration Act was signed into law (U.S. Public Law 102-495), authorizing the Department of the Interior to acquire the Elwha and Glines Canyon dams for the purpose of restoring the Elwha River ecosystem and anadromous fish populations [13]. Due to the volume of sediment, a three-year staged removal was recommended to balance the severity and duration of downstream sediment impacts [8,14]. Removal of both dams began in September 2011 and ended in August 2014 (Fig 2). During the first year of dam removal (water year 2012; WY = October - September), the combined quantity of sediment transported downstream from Lake Aldwell (1.1 Mt) and Lake Mills (0.19 Mt) was three to four times greater than the long-term average annual sediment supply [4,5]. Following complete removal of the Elwha Dam in March 2012, downstream suspended sediment concentrations (SSC) reached 6,500 mg/L in the spring of 2012 and exceeded 1,000 mg/L for 9% of the year (Fig 2C).

Peak sediment transport occurred in WY 2013, when coarse sediment from Lake Mills began spilling over Glines Canyon Dam in Oct 2012. Although active dam removal was suspended for most of 2013, over a third of the remaining sediment in Lake Mills (~ 8.8 Mt) was transported downstream during this time period, along with an additional 0.22 Mt from Lake Aldwell [5]. This combined 9.1 Mt represented roughly 20 times the river's average annual sediment supply, resulting in prolonged periods of elevated water turbidity. Average daily SSC exceeded 1,000 mg/L for over half of the year, and peaked at over 10,000 mg/L during high flow events [19].

In the final year of active dam removal (WY 2014), deconstruction of the remaining 16 m of Glines Canyon Dam resumed in October 2013, with the final blast occurring in August 2014. During WY 2014, 3.5 Mt (Mills) and 0.30 Mt (Aldwell) of sediment was transported from the two reservoirs. While less than WY 2013, SSC was still over eight times larger than average background levels, exceeding 1,000 mg/L for 30% of the year, and peaking at 19,755 mg/L in the spring of 2014.

Our study focused on the short-term sediment impacts to aquatic food webs during the three years of active dam removal (WYs 2012–2014). Many studies have investigated the effects of dam removal on benthic invertebrates (reviewed by [20]), and other predominantly

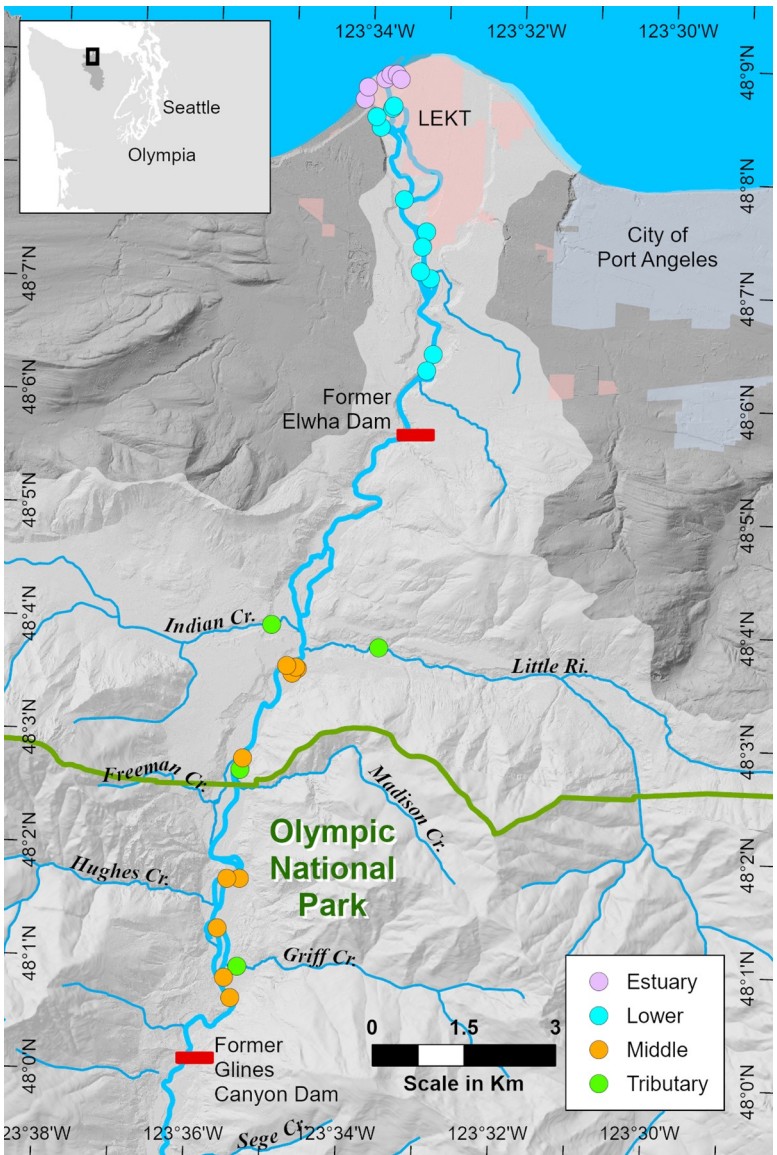

**Fig 1. Study region.** Distribution of samples sites across three sediment-impacted (Estuary, Lower Elwha, Middle Elwha) and one control (Tributary) section, relative to former dams on the Elwha River on the Olympic Peninsula (inset map) of Washington State, USA. Estuary markers represent sample areas with multiple replicates. Reservation lands of the Lower Elwha Klallam Tribe (LEKT) are shown in pale pink; the horizontal green line indicates the Olympic National Park Boundary. Map scale 1:124,000. Data layers: LiDaR DEM hillshade: USGS, LEKT, NOAA; Washington State DNR: tributaries and drainage basin; City of Port Angeles, NPS, LEKT: political boundaries. Generated by Randall E. McCoy, Lower Elwha Klallam Tribe Department of Natural Resources, on August 19, 2020 using ArcGIS Pro [GIS]. Version 2.5.2. Redlands, CA: ESRI, 2019. File: Elwha River Watershed 19 (fish diet figure). aprx.

laboratory-based investigations have examined the effects of increased turbidity and bed disturbance on salmonid foraging [21–23]. However, we are not aware of any published studies that have investigated the coupled effects of dam removal on prey availability and prey consumption.

We examined how prey availability and diet composition of juvenile salmonids (*Oncorhynchus* spp.) in the Elwha River responded to elevated sedimentation during dam removal.

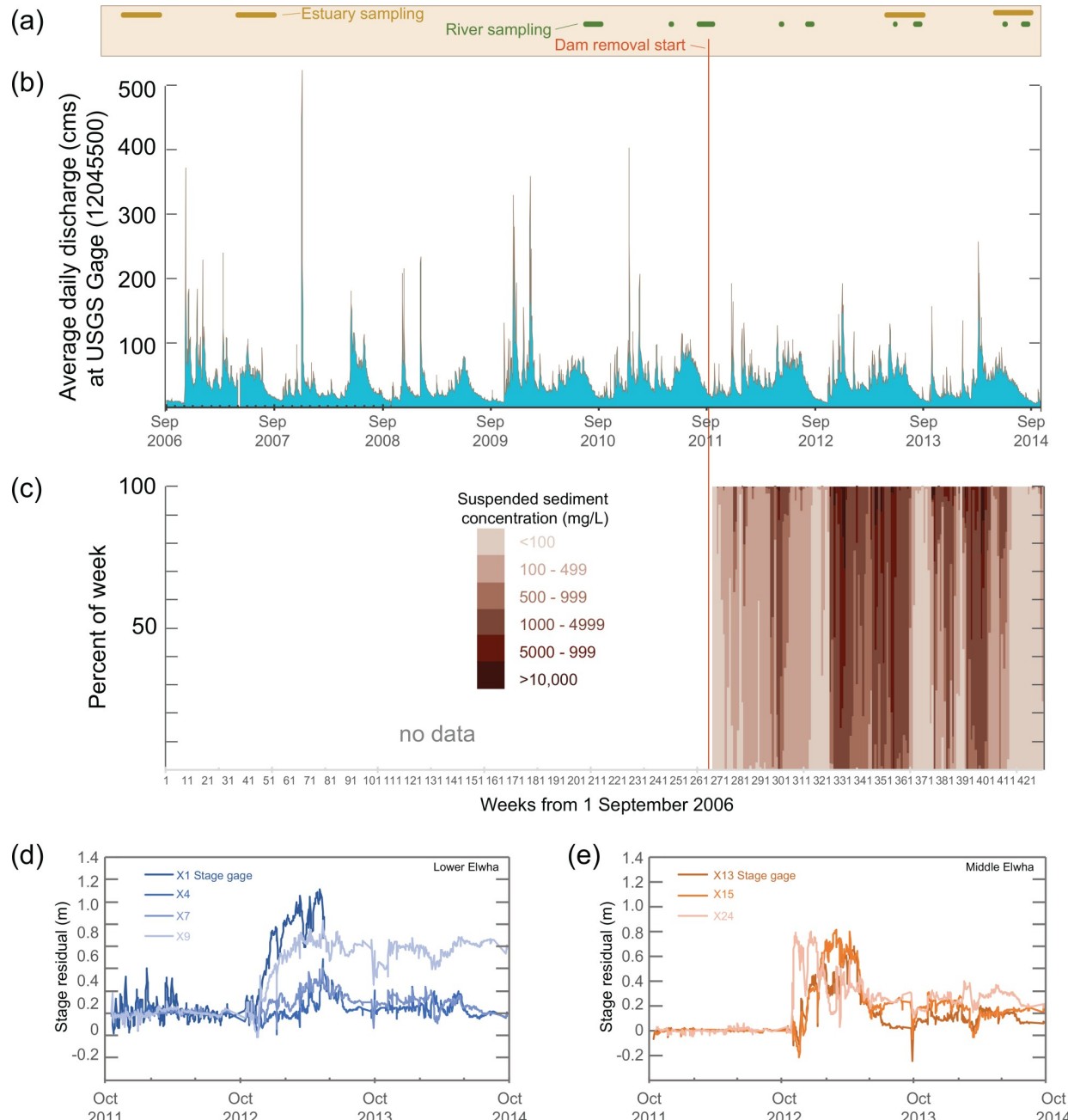

**Fig 2. Timeline of Elwha River dam removals.** Dates are shown relative to (a) Sampling schedule for fish and invertebrate collections, (b) average daily discharge measured at U.S. Gological Survey (USGS) McDonald Bridge Gage (#12045500, [15]) during water years WY2006 - WY2014, (c) suspended sediment concentration histogram [16] showing the percent of weeks spent in six different concentration bins based on 15-minute turbidity measurements downstream of both dams [5], and a stage-discharge residual analysis based on Anderson and Konrad [17] showing how sediment release from dam removal changed bed elevations across a network of river stage recorders operated by Bountry et al [18] in (d) the Lower and (e) Middle Elwha River. The number of each stage recorder increases in an upstream direction (i.e., X1 closest to the river's mouth).

Specifically, we examined how invertebrate prey densities and composition, fish diet, and prey selectivity changed spatially (relative to former dam locations and reservoirs), and temporally (seasonally, and relative to timing of sediment releases). Because the restoration of

anadromous fish communities was a primary goal of the project, it is important to understand how the short-term effects of dam removal impacted the prey base, selectivity, and diet of juvenile salmon. We connect observed food web response to likely mechanisms driving change and discuss the ramifications for long-term recovery of the Elwha River.

## Methods

### Study area

Scientific collection permits for this study were obtained prior to field collections, and were issued by the National Marine Fisheries Service (Tribal Plan Limit 50 CFR 223.204), Washington Department of Fish and Wildlife (Permit# 10-248, 12-257, 13-257, 14-058a), U.S. Fish and Wildlife Service (Permit# TE-702631, sub-permit FWSWFWO-09 to FWSWFWO-14), and by Olympic National Park (Permit# OLYM-2011-SCI-0049, OLYM-2012-SCI-0021, OLYM-2013-SCI-0041, OLYM-2014-SCI-0041). All sampling and handling of fish and invertebrates was carried out in accordance with permit conditions.

At 72 km in length, the Elwha is a short, steep river that originates in the Olympic Mountains and flows northward to empty into the Strait of Juan de Fuca 9 km west of the city of Port Angeles, Washington (Fig 1). Eighty-three percent of the 833 km$^2$ river basin is protected within the relatively undisturbed coniferous forests and designated wilderness of Olympic National Park. Roughly 14 km of the Elwha is downstream of the park boundary and flows through second-growth forests on a patchwork of private, state, and tribal lands. With heavy precipitation (annual average = 143 cm), past glacial activity and active tectonics, the upper Elwha River has a relatively high annual sediment load of 0.34 ± 0.08 Mt [19]. The mean annual flow (1918–2010) at USGS gauge 12045500 is 43 m$^3$/s, with seasonal peaks in discharge occurring during the rainy season from fall and early winter, and again during spring snowmelt [24] (Fig 2B).

Although the dams blocked anadromous fish passage to over 90% of the watershed, the lower 7.9 km of river downstream of the Elwha Dam still supported populations of five anadromous salmon species (Chinook (*O. tshawytscha*), chum (*O. keta*), coho (*O. kisutch*), pink (*O. gorbuscha*), sockeye (*O. nerka*)), as well as anadromous steelhead (*O. mykiss*), cutthroat (*O. clarkii*), bull trout (*Salvelinus confluentus*), Pacific lamprey (*Entosphenus tridentatus*), and eulachon (*Thaleichthys pacificus*) [7]. Resident fish species found in the river basin included bull trout, rainbow trout, sculpin (*Cottus* spp.), threespine stickleback (*Gasterosteus aculeatus*), redside shiner (*Richardsonius balteatus*), surf smelt (*Hypomesus pretiosus*), and non-native brook trout (*S. fontinalis*) [11]. Coho and steelhead populations are augmented by a tribal fish hatchery at rkm 3, and the Chinook population by a state fish hatchery at rkm 5.

### Experimental design

We merged two independent field studies conducted in estuarine and river habitats. Although sample frequency and collection methods were not identical, both studies shared a common objective to examine the response of prey availability and salmonid diet to dam removal. Both field efforts include data collected before and during dam removal over different habitat types and seasons. Prey availability at estuarine sites was evaluated by sampling benthic and shoreline invertebrates, and at river sites by sampling invertebrate drift. Both studies evaluated fish diet via non-lethal gastric lavage at the same locations invertebrate prey were collected.

We collected samples from three sediment-impacted and one reference section (Fig 1). Impacted sections included the estuary (elevation 2 m) and two river sections: the lower Elwha River extending from the estuary to the former Elwha Dam (7 rkm, elevations 3–25 m, hereafter referred to as LE); and the middle Elwha (ME) from upstream of the Lake Aldwell inlet to

**Table 1. Number of invertebrate and fish diet samples collected in the estuary and river before (Pre) and during (2012-2014) dam removal.**

| Sample | Section | Pre Spring | Pre Summer | 2012 Spring | 2012 Summer | 2013 Spring | 2013 Summer | 2014[2] Spring | 2014[2] Summer |
|---|---|---|---|---|---|---|---|---|---|
| Benthic | Estuary | 18 | 18 | — | — | 18 | 18 | — | — |
| Fallout | Estuary | 9 | 10 | — | — | 10 | 10 | — | — |
| Drift | LE | 3 | 8 | 2 | 9 | 2 | 8 | 3 | 7 |
| Drift | ME | 3 | 6 | 2 | 6 | 2 | 8 | 3 | 8 |
| Drift | TR | 1 | 4 | 4 | 4 | 4 | 4 | 4 | 4 |
| Fish | Estuary[1] | 60 | 56 | — | — | 73 | 53 | 62 | 88 |
| Fish | LE | 29 | 99 | 19 | 57 | 3 | 8 | 4 | 17 |
| Fish | ME | 29 | 69 | 18 | 53 | 15 | 7 | 10 | 17 |
| Fish | TR | 9 | 49 | 34 | 39 | 34 | 33 | 30 | 25 |

Invertebrate samples represent the number of unique sites sampled across each section and year, whereas fish diet numbers represent the total number of individual fish sampled in each section and year. LE = downstream of former Elwha Dam, ME = between the former Elwha Dam and Glines Canyon Dam, TR = tributary sites; see also Fig 1.

[1] A portion of the estuary diet samples from 2007 was collected by another researcher, as reported by Shaffer et al. (2009).

[2] All spring 2014 river samples were lost in transit by UPS; no taxonomic data available for this time period.

the former Glines Canyon Dam (12 km, elevations 62–110 m). Four tributaries (TR) that flowed into ME (Little River, Indian River, Madison Creek, Griff Creek; elevations 68–112 m) served as a reference section as they were free from sediment impacts associated with dam removal. However, tributary sites did experience other changes during dam removal (see Discussion).

Across all sections, we sampled 28 unique locations in the estuary and 24 in the river and tributaries (Fig 1). River sites included two habitat types: mainstem and floodplain channels. The distribution of sites was non-random, but dictated by access and overlap with long-term monitoring stations [12,24–26]. The total number of sites sampled in a given year and season changed over time due to variation in year-to-year river discharge and a rapidly shifting habitat mosaic during dam removal (Table 1). In the river, high flows precluded sampling mainstem sites in spring. Due to this smaller sample size, we combined LE and ME sites into a pooled LME category in spring to contrast with non-sediment impacted tributaries.

The timing and frequency of data collection differed between estuary and river sections (Fig 2A). All sections were sampled both in spring (May–June) and summer (July–August). In the estuary, we sampled fish diet before dam removal in 2006–2007 and during dam removal in 2013–2014. Estuary invertebrate sampling occurred in 2007 and 2013. For river and tributary sections, we sampled invertebrates and fish before dam removal in 2010–2011 and during dam removal from 2012–2014. Due to uneven sample effort in pre-dam removal years, we grouped all data collected prior to the start of dam removal into one "Pre" dam removal category for purposes of analyses.

## Data collection

We collected benthic and terrestrial invertebrates in the estuary using two different techniques. We used a petite Ponar grab to collect surface benthic invertebrates from a 15 x 15 cm area. Sediment grabs were sorted through a 500-μm sieve, fixed in formalin for 3 to 5 days, and transferred to ethanol until processing. Estuary shoreline invertebrate samples were collected in fallout trap arrays: five clear plastic bins filled with approximately 5 cm of soapy water and deployed near the estuary shoreline for three days in emergent and shrub vegetation. Bin

contents were filtered through a fine sieve and invertebrates stored in ethanol until processing (additional sampling details in [27]). Shoreline terrestrial invertebrate samples were collected in the east estuary only.

In riverine habitat, we collected both aquatic and terrestrial invertebrates via drift sampling [28]. Two 250-μm-mesh, 0.14 $m^2$-frame drift nets were placed side-by-side at the upstream end of each sample reach, directed perpendicular to flow, and secured above the stream bottom with rebar. We left nets in place over a timed 30–60-minute interval and measured adjacent water velocity and depth to calculate the total water volume sampled. Invertebrates captured in the nets were rinsed through a 200-μm-mesh soil sieve and combined into one sample preserved in ethanol.

We used different techniques to capture juvenile salmonids from the river and estuary and focused on different species for diet analyses. To collect fish in deep slow-moving estuarine habitats, we used standard Puget Sound beach seining protocols, deploying a seine from bank to bank by small skiff [29]. Coho (*O. kisutch*) and Chinook (*O. tshawytscha*) were the numerically dominant salmonids captured at estuarine sites. Estuary fish abundance data from this sampling effort are summarized in [26]; we report here on their diet contents.

Because there were no anadromous fish present upstream of the Elwha Dam prior to dam removal, our river sampling focused on resident trout (*O. mykiss*, except for *O. clarkii* at one tributary site). In shallow (< 1.5 m) fast-flowing riverine habitats, we used a Smith-Root backpack electrofisher to collect salmonids within a 500-m sample reach. Multiple passes were made until either 10 trout were captured, or a period of 90 minutes had elapsed. Juvenile salmonid sampling in rivers was not designed to estimate fish densities, only to collect specimens for diet analyses.

Stomach contents from a sub-sample of target salmonid species were extracted via nonlethal gastric lavage and preserved in ethanol [27,30]. At both river and estuary sites, captured fish were transferred to buckets containing aerated ambient water and kept cool until handling. We anesthetized fish in a diluted solution of tricaine methanesulfonate (MS-222) to count, identify, and measure fork length (FL) to the nearest mm and weight to the nearest 0.1 g. When possible, stomach contents from ten individuals of each species between 55–199 mm FL were collected at each site during each sampling event. Due to accidental electrofishing mortality (0.03%), we also collected diet samples from six smaller fish (42–54 mm FL). Fish with no regurgitated prey were recorded as empty stomachs. After recovery from anesthesia, all fish were released at the point of capture.

Fish diet and invertebrate samples were sent to a professional taxonomist for processing and identification to the lowest practical taxonomic level. Taxa were classified by their origin: terrestrial or aquatic (invertebrates that are aquatic in every life stage as well as the terrestrial life stages of aquatic larvae). Aquatic taxa were typically identified to the genus or species level (S1 Table) and terrestrial taxa to family or genus (S2 Table). Some less-common taxa or partially-digested prey items were identified to Order or Class. Invertebrate density was calculated using surface area ($m^2$) for estuarine samples, and water volume ($m^3$) for river drift samples.

Body parameter measurements were recorded for all invertebrate taxa to the nearest mm, either as head-capsule width or total length. From these measurements, we calculated the energy content (joules) of each invertebrate based on regression equations for converting body measurements to biomass (S3 Table), and then from biomass to energy (S4 Table). Regression equations came from the literature, or were developed from individuals collected during this study. These calculations were life-stage specific, and made at the family-level or next available taxonomic level reported in the literature. Individual values were summed across samples to calculate invertebrate density both numerically and energetically.

## Sample metrics

We calculated invertebrate and fish diet metrics for each unique sample event at every site (hereafter referred to as "site sample event"). We measured invertebrate density in terms of numeric (No·m$^{-2}$ (estuary)/No·m$^{-3}$ (river)) and energetic (KJ·m$^{-2}$ (estuary)/J·m$^{-3}$ (river)) density. We also calculated the taxa richness of each sample (number of unique taxa present) and the proportion of terrestrial individuals. In the estuary, benthic and fallout prey availability data were analyzed separately due to the different habitats and collection methods. As multiple fish were sampled on each site sample event, we based all statistical analyses on means. On a per fish basis, we calculated total prey energy (J), number of prey items, and number of unique prey items (taxa richness) in each stomach. For prey origin, we calculated the proportion of each diet sample composed of terrestrial-origin taxa.

In order to quantify overlap between fish diet and invertebrate prey availability, for each site sample event we calculated Bray-Curtis similarity coefficients between the paired biological assemblages [31]. Values ranged from no overlap (0) to 100 (perfect similarity) [32]. As units varied between the two datasets, we standardized all data prior to analyses.

We compared invertebrate prey availability to diet composition using a modified Index of Relative Importance (IRI) approach, substituting prey energy content (joules) for volume [33,34]. Treating each section-season-year group separately, we calculated IRI values for every invertebrate order present in the environment and in diet samples as: IRI = F (N + J); F = frequency of occurrence percentage, N = numerical percentage, J = energetic percentage. To compare across multiple groups, we standardized all IRI values by group total such that % IRI values ranged from 0–100. To visualize shifts in prey selectivity over time, we plotted environmental %IRI versus diet %IRI in two dimensional space. Taxa that plot above the 1:1 line indicate selection by fish.

## Statistical analyses

To evaluate dam removal effects on univariate metrics (density, taxa richness, percentage of terrestrial individuals, Bray-Curtis coefficients) we tested for differences using a one-way analysis of variance (ANOVA) for estuary data to examine differences between years and a two-way ANOVA for river data to examine the interaction between year and section. All data were transformed as appropriate to meet test assumptions. If main terms were significant, we tested for pair-wise year differences by section with Tukey's Honest Significant Difference (HSD) test at $\alpha$ = 0.05. Because spring river sampling did not include mainstem sites, we analyzed seasons separately. For summer data, we report the interaction between Habitat Type (mainstem or floodplain) and year for LE and ME.

To examine multivariate differences in invertebrate and fish diet assemblage structure, we used a suite of complementary techniques available in the statistical software packages PRIMER (version 7.0.13) and PERMANOVA (version 1.0.3). To balance the contributions of common and rare taxa, we square-root (drift) or fourth-root (all other datasets) transformed our data and created Bray-Curtis similarity matrices. To account for empty or near-empty samples, we added a dummy species with an abundance set to 0.1 to all samples, thus avoiding undefined Bray-Curtis coefficients. To test for differences in diet by size class in the river (young-of-year [hereafter YOY] versus fish older than a year [hereafter 1+]) and for differences by species (Chinook versus coho) in the estuary, we applied analysis of similarity (ANOSIM), a non-parametric analog to ANOVA that does not require balanced replication or homogeneity of variance [32].

We tested for larger-scale patterns in invertebrate and fish diet relative to section and year using PERMANOVA—a permutation-based method to test for compositional differences

among groups of sites based on resemblance measures [35]. For estuary datasets we used a one-way design with **y**ear as a fixed factor; for riverine datasets we used a crossed design with **y**ear and section as fixed factors. We used Type III partial-sums of squares and permutation of residuals under a reduced model.

If the main terms (**y**ear, and **y**ear x section for river) were significant, we tested for pairwise differences by **y**ear. To adjust for multiple tests across the three river sections, we applied Bonferroni corrections. For tests with fewer than 40 possible unique permutations, we report Monte Carlo asymptotic *P*-values. Where significant differences were detected by **y**ear, we used the similarity percentage (SIMPER) routine in PRIMER to determine which taxa contributed most to differences between **y**ear-section groups.

To visualize changes in the multivariate taxonomic structure of invertebrate prey and fish diet over time, we used non-metric multi-dimensional scaling (nMDS) and examined each section and season individually. In order to visualize the relative size and direction of different components of variation in our overall experimental design, we created Bray-Curtis resemblance matrices based on the distance between data centroids by **y**ear, section, and season. We then used nMDS to plot all section centroids together and show their varied trajectories over time.

## Results

### Estuary prey availability

We observed 40 unique aquatic-origin taxa in the estuary benthic invertebrate samples (S1 Table), with mean numeric density ranging from 4111–4561 individuals m$^{-2}$ across seasons prior to dam removal and 1829–2309 individuals m$^{-2}$ during dam removal—a decline of greater than 50% (Table 2). Benthic density and taxa richness were significantly lower during

**Table 2. Univariate metrics (mean ± 1 SD) from estuary benthic and fallout invertebrate samples.**

| Sample | Season | Pre | 2013 |
|---|---|---|---|
| **Numeric Density (No./m$^2$)** | | | |
| Benthic | Spring | 4111 (± 2960)[a] | 1829 (± 2323)[b] |
| Benthic | Summer | 4561 (± 3059)[a] | 2309 (± 3131)[b] |
| Fallout | Spring | 692 (± 207)[a] | 99 (± 50)[b] |
| Fallout | Summer | 622 (±229) | 656 (± 363) |
| **Taxa Richness (Number of unique taxa)** | | | |
| Benthic | Spring | 6.72 (± 1.88)[a] | 4.33 (± 2.33)[b] |
| Benthic | Summer | 7.89 (± 2.58)[a] | 4.39 (± 1.89)[b] |
| Fallout | Spring | 20.61 (± 4.32)[a] | 11.5 (± 5.55)[b] |
| Fallout | Summer | 22.8 (± 4.99) | 21.7 (± 8.50) |
| **Terrestrial Percentage (%)** | | | |
| Benthic | Spring | 0 | 0 |
| Benthic | Summer | 0 | 0 |
| Fallout | Spring | 23.5 (± 13.4)[a] | 60.0 (± 24.1)[b] |
| Fallout | Summer | 33.9 (± 13.9) | 34.2 (± 27.0) |
| **Energy Density (KJ/m$^2$)** | | | |
| Benthic | Spring | 130.19 (± 183.69) | 135.40 (± 224.28) |
| Benthic | Summer | 163.80 (± 204.16) | 189.13 (± 332.95) |
| Fallout | Spring | 4.96 (± 1.46)[a] | 1.01 (± 0.54)[b] |
| Fallout | Summer | 5.65 (± 1.66) | 4.89 (± 3.08) |

Different letters indicate significant (*P* < 0.05) differences between years based on 1-way ANOVA.

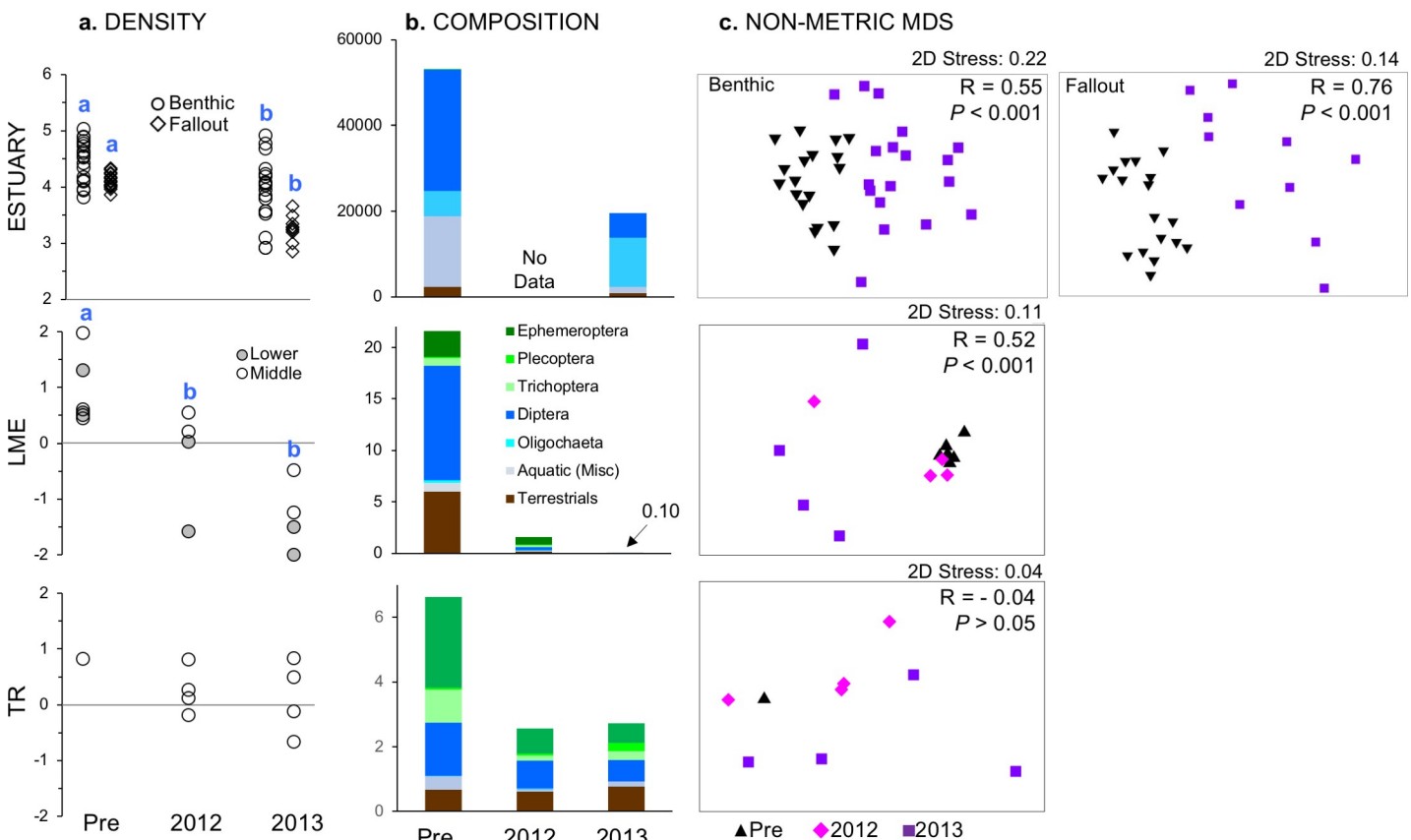

**Fig 3. Spring invertebrate density by section and sample year.** Data are plotted as (a) log density by site, (b) composition by major taxonomic groups (section means), and (c) nMDS, with global sample statistic and significance for one-way ANOSIM by year; note that ordinations with 2D stress values > 0.2 may not adequately represent data structure [36]. For panel a, different letters indicate significant year differences ($P < 0.05$) within a section based on Tukey's HSD. Note that y-axis range varies by section, and that collection methods varied between estuary and river sections. All data are based on numerical density: counts per m$^2$ for estuary and per m$^3$ for all other sections. For estuary data, composition data in panel b is combined for benthic and fallout samples.

dam removal than before, in both spring and summer (1-way ANOVA, $P < 0.05$) (Figs 3A and 4A). Amphipods and ostracods were among the most abundant taxa in benthic samples prior to dam removal, but were completely absent during dam removal (see Aquatic (Misc) category in Figs 3B and 4B). Dipterans were still abundant during dam removal but their density declined; Oligochaetes, on the other hand, increased in density during dam removal.

We identified 124 unique taxa of invertebrates captured in estuary shoreline fallout traps (53:71 aquatic:terrestrial origin; S1 and S2 Tables), with average densities ranging from 622–692 individuals m$^{-2}$ across seasons prior to dam removal and 99–656 individuals m$^{-2}$ following dam removal (Table 2). Fallout invertebrate density and taxa richness were significantly lower in spring during dam removal than before (1-way ANOVA, $P < 0.05$), but there was no difference in summer ($P > 0.05$) (Table 2; Figs 3A and 5A). The proportion of terrestrial invertebrates in fallout samples tripled during dam removal in the spring, but was not significantly different in the summer ($P < 0.05$).

Benthic taxonomic composition varied significantly by year during both spring and summer (PERMANOVA, $P < 0.001$; Table 3; Figs 3C and 6C). In the spring, ten taxa contributed at least 2% of the average dissimilarity between years, including the near complete loss during dam removal of Ostracods, the Amphipod family Corophiidae, as well as damselfly and

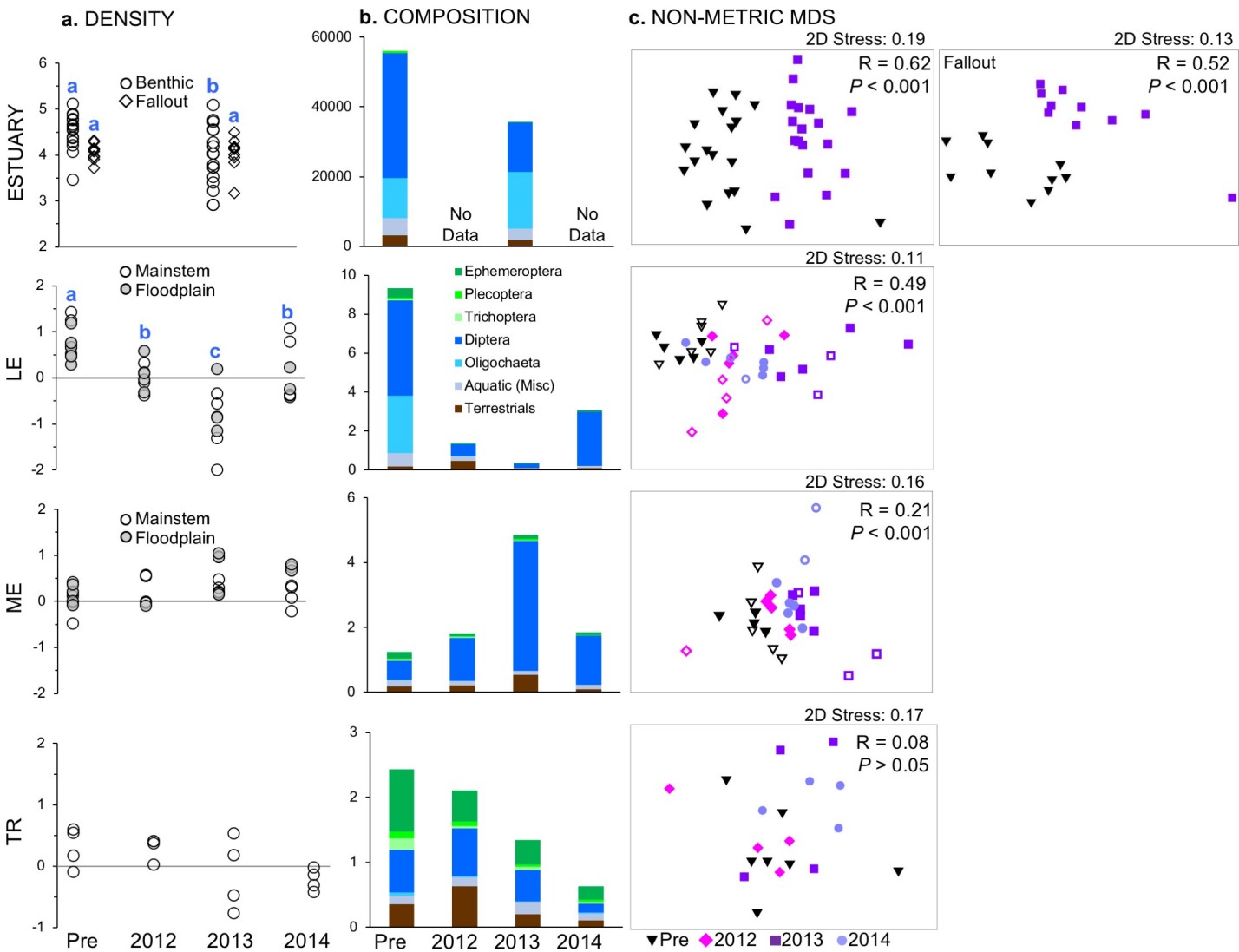

**Fig 4. Summer invertebrate density by section and sample year.** Data are plotted as (a) log density by site, (b) composition by major taxonomic groups (section means), and (c) nMDS, with global sample statistic and significance for one-way ANOSIM by year. For panel a, different letters indicate significant year differences (*P* < 0.05) within a section based on Tukey's HSD. Note that y-axis range varies by section, and that collection methods varied between estuary and river sections. For panel c, floodplain sites are indicated with outlined symbols. All data are based on numerical density: counts per m² for estuary and per m³ for all other sections. For estuary data, composition data in panel b is combined for benthic and fallout samples.

caddisfly families (S1A Fig). An additional four taxa disappeared from benthic samples during dam removal summer seasons, while Oligochaetes maintained or increased in density (S1B Fig).

Fallout invertebrate taxonomic composition also varied significantly by year in spring and summer seasons (PERMANOVA, *P* < 0.001; Table 3; Figs 3C and 7C). The SIMPER analysis showed that 19 taxa in the spring and 13 taxa in the summer contributed at least 2% of the average dissimilarity between years, suggesting that invertebrates sampled in fallout traps changed more broadly than those in the benthic community. In the spring, 11 of the 19 taxa declined during dam removal compared to before removal (S1C Fig), while the abundances of summer taxa predominantly stayed the same or increased during dam removal (S1D Fig).

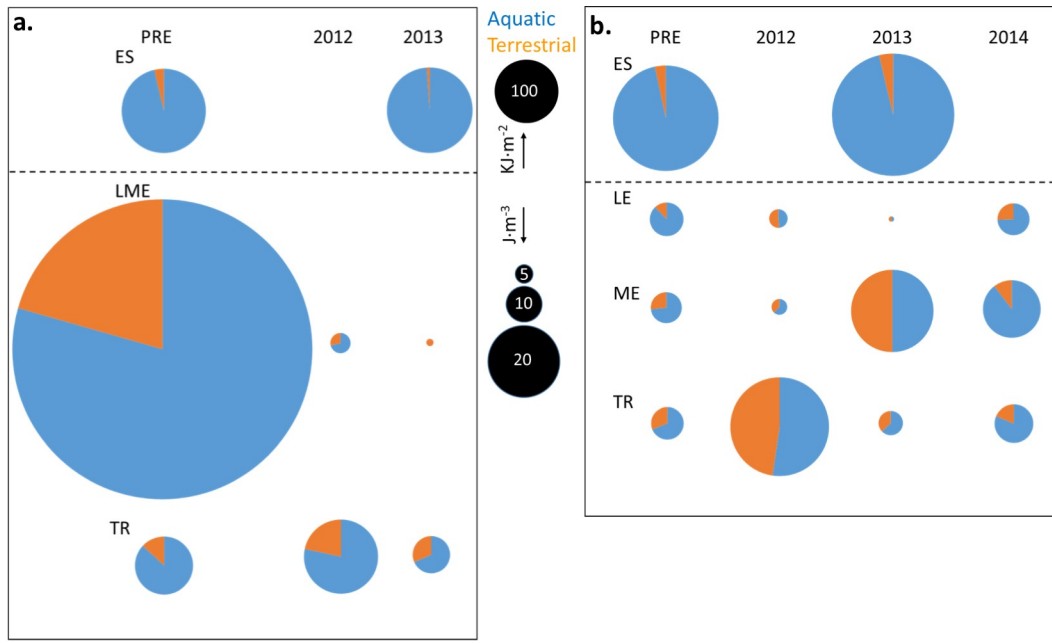

**Fig 5. Mean invertebrate energy density by habitat of origin (aquatic or terrestrial).** Data are plotted by section and year for (a) spring and (b) summer. Note that density units are expressed as KJ·m$^{-2}$ in the estuary, but as J·m$^{-3}$ in river sections.

When we converted numeric invertebrate density into energetic density, there were no differences before and during dam removal for benthic samples in either spring or summer (1-way ANOVA, $P > 0.05$; Table 2). For fallout samples, energy density was significantly lower in spring during dam removal (1-way ANOVA, $P < 0.05$; Table 2), but there was no difference in summer ($P > 0.05$). Total energy density of combined benthic and fallout samples was similar before and during dam removal, both in spring and summer (Fig 5).

## River prey availability

We observed 295 unique invertebrate taxa in drift samples (60:40 ratio of aquatic:terrestrial origin; S1 and S2 Tables). Most taxa were insects (83%), with the remainder consisting of arachnids, crustaceans, collembola, molluscs, worms, myriapoda, and hydrozoa. Although the

**Table 3. PERMANOVA results for main test of year effect on estuary benthic and fallout invertebrates.**

| Season | Sample | Source | df | SS (III) | ECV | Perm | Pseudo-*F* | *P* |
|---|---|---|---|---|---|---|---|---|
| Spring | Benthic | Year | 1 | 15298 | 27.763 | 999 | 10.737 | **0.001** |
|  | Benthic | Residual | 34 | 48443 | 1424.8 |  |  |  |
|  | Fallout | Year | 1 | 11156 | 27.611 | 998 | 83234 | **0.001** |
|  | Fallout | Residual | 26 | 35228 | 36.809 |  |  |  |
| Summer | Benthic | Year | 1 | 13701 | 26.464 | 998 | 12.515 | **0.001** |
|  | Benthic | Residual | 34 | 37223 | 1094.8 |  |  |  |
|  | Fallout | Year | 1 | 7546 | 24.785 | 995 | 5.379 | **0.001** |
|  | Fallout | Residual | 18 | 25248 | 37.452 |  |  |  |

Test based on partial sums of squares (SS) and permutation of residuals under a reduced model. Perm = total number of unique possible permutations, df = denominator degrees of freedom, and *P*-values are based on permutations. ECV = square root of the estimated components of variation in the model. Pseudo-F values are the multivariate analog to the univariate F statistic. Taxonomic resolution is at the family level for fourth-root transformed invertebrate density. Bolded black values indicate $P < 0.05$.

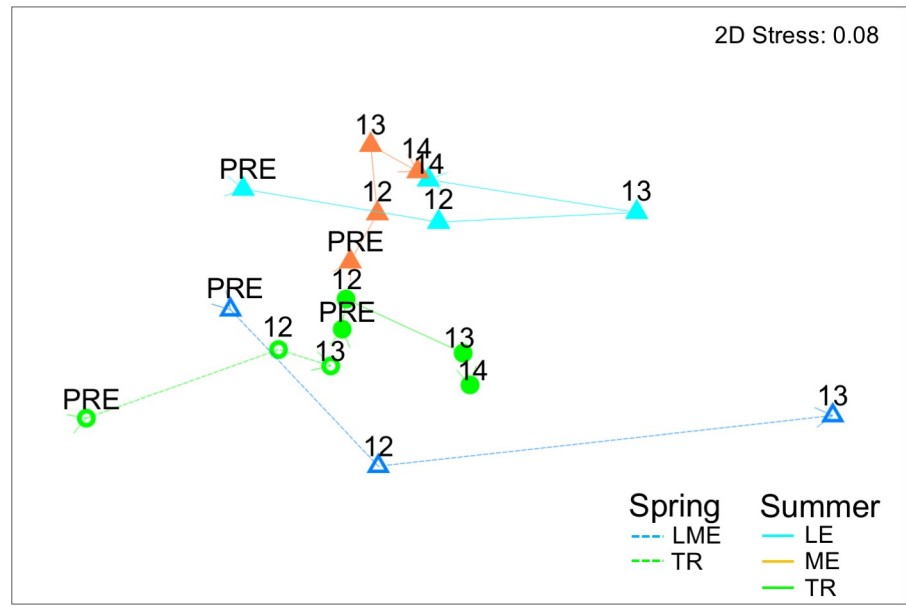

**Fig 6. Invertebrate drift nMDS plot of data centroids.** Trajectories showing relative direction and magnitude of change by season and river section over time.

proportion of drift from terrestrial sources was relatively small prior to dam removal (averaging 5% in the spring and 10% in the summer), terrestrial-origin taxa contributed a third of total taxa richness across all river sections during dam removal—despite coarser taxonomic resolution than aquatic taxa.

Drift density, taxa richness, and terrestrial percentage all significantly changed during dam removal in one or more sediment-impacted sections of the river, but not in reference section TR (Table 4). We detected a significant **y**ear effect on all three metrics in both spring and summer (2-way ANOVA, $P < 0.05$). There was a significant interaction of **y**ear x section on density and terrestrials in summer, but not in spring. We did not observe an interaction effect on taxa richness in either season, or an effect of Habitat Type (mainstem versus floodplain) on any of the three metrics (**y**ear x habitat, $P > 0.05$).

Drift density and taxa richness declined in most impacted sections during dam removal, while terrestrial percentage increased. Spring drift density in LME declined by 93% in 2012 and by 99.9% in 2013 to only 0.10 ·m$^{-2}$ (Tukey's HSD, $P < 0.05$) (Fig 3A). Spring LME taxa richness decreased over an order of magnitude, and was reduced to a mean of two taxa by 2013 ($P < 0.05$; Table 4). Conversely, the percentage of terrestrial invertebrates in LME rose from 12% before dam removal to 80% by 2013 ($P < 0.05$; Fig 3A; Table 4).

We observed the same patterns in summer for LE (Fig 3B; Table 4). Mean drift density declined 96% during dam removal to a low of 0.33 ·m$^{-2}$ in 2013. Taxa richness also significantly decreased in LE in 2012 and 2013 ($P < 0.05$), but more taxa persisted than did in spring. The percentage of drift originating from the terrestrial zone increased ten-fold during these two years ($P < 0.05$), but remained below 30%. In 2014, values for all three metrics neared levels observed before dam removal. We did not detect a significant change in ME in either density or the percentage of terrestrials, but taxa richness was significantly lower in 2014 than before dam removal ($P < 0.05$).

Overall multivariate taxonomic composition of the drift changed across all sediment-impacted sections during dam removal, but again not in reference TR (Table 5A). We detected

a significant **y**ear effect in both seasons, and a significant **y**ear x section interaction in summer (PERMANOVA two-way crossed design; $P < 0.01$). While spring density of nearly all LME drift taxa decreased in 2012 (S2A Fig), no families disappeared altogether and composition did not significantly differ from before dam removal ($P = 0.07$; Table 5B). That changed in spring 2013 ($P < 0.01$), when the only aquatic taxa present in our drift samples were the riffle beetle Elmidae and the alderfly Sialidae.

In summer, taxonomic composition in LE differed between every year pair except 2012/ 2014 (Table 5B). Beginning in 2012, we observed widespread changes across almost all taxonomic groups (S2B Fig). Density decreased for 87% of aquatic families but increased for 74% of terrestrial ones. By 2013, over 85% of both aquatic and terrestrial families had decreased or disappeared altogether. Although total drift density increased in 2014, taxonomic composition was still markedly different than before dam removal. Twenty-six aquatic families present before dam removal were absent in 2014, including all seven Trichoptera families and five of six Plecoptera families. The biggest increases in 2014 were by Dipterans, which comprised 91% of all individuals in 2014 compared to 53% before dam removal (Fig 4B). Conversely, aquatic oligochaetes comprised 0.33% of drift in 2014 compared to 32% before dam removal.

In ME, we detected taxonomic changes in the summers of 2013 and 2014 relative to both pre-dam removal and to 2012 (Table 5B). Differences between years were driven by many of

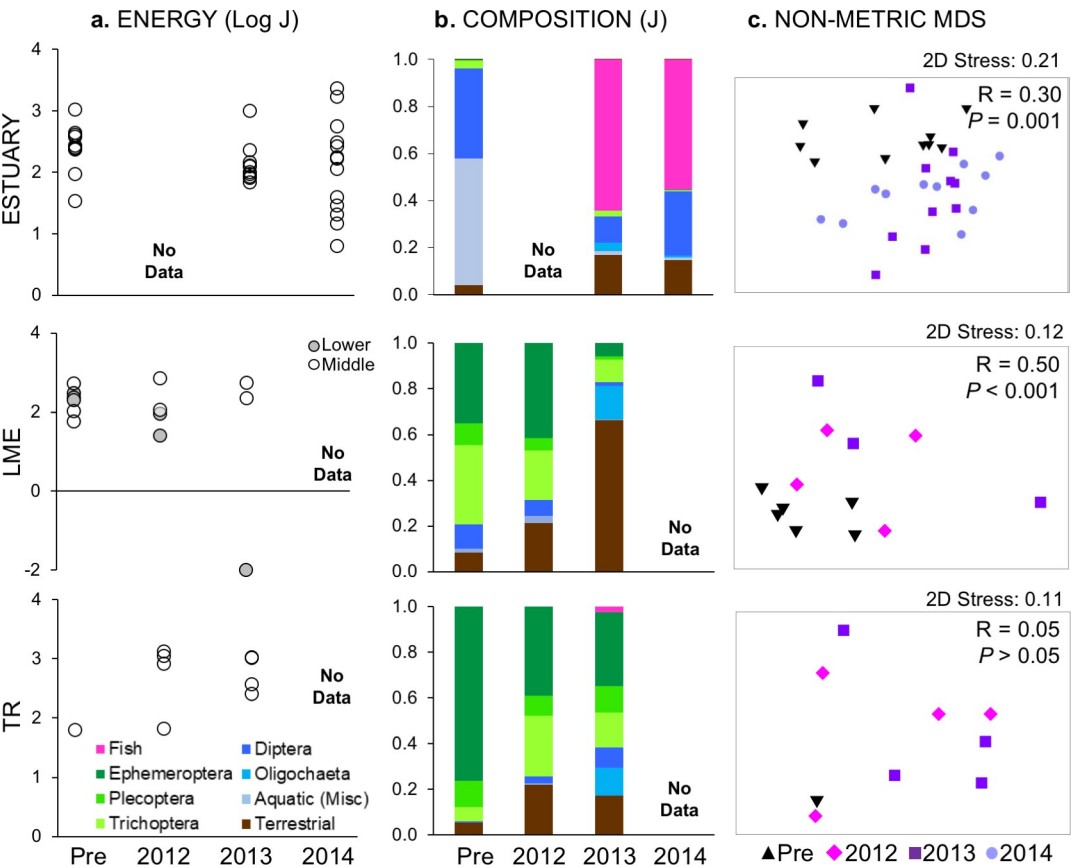

**Fig 7. Spring fish diet by section and sample year.** Data are plotted as (a) mean total prey energy per fish by site, (b) relative composition by major taxonomic groups (section means), and (c) nMDS, with global sample statistic and significance for one-way ANOSIM by year; note that ordinations with 2D stress values > 0.2 may not adequately represent data structure [36]. All data are in J. For panel a, different letters indicate significant year differences ($P < 0.05$) within a section. Note that y-axis range varies by section, and that collection methods varied between estuary and river sections.

**Table 4. Univariate metrics (mean values ± 1 SD) from river invertebrates collected by drift sampling.**

| Season | Section | Pre | 2012 | 2013 | 2014 |
|---|---|---|---|---|---|
| | | **Numeric Density (No./m$^3$)** | | | |
| Spring | LME | 21.60 (± 36.93)[a] | 1.56 (± 1.48)[b] | 0.10 (± 0.15)[b] | — |
| | TR | 6.63 (± NA) | 2.57 (± 2.65) | 2.72 (± 3.00) | — |
| Summer | LE | 9.27 (± 8.43)[a] | 1.36 (± 1.03)[b] | 0.33 (± 0.52)[c] | 3.07 (± 4.43)[b] |
| | ME | 1.33 (± 0.66) | 3.18 (± 3.84) | 4.86 (± 4.15) | 1.85 (± 1.74) |
| | TR | 1.38 (± 1.64) | 2.11 (± 0.71) | 1.34 (± 1.49) | 0.63 (± 0.26) |
| | | **Taxa Richness (Number of unique taxa)** | | | |
| Spring | LME | 31.83 (± 11.60)[a] | 20.00 (± 19.92)[ab] | 2.00 (± 2.16)[b] | — |
| | TR | 42.00 (± NA) | 35.25 (± 12.34) | 26.00 (± 9.87) | — |
| Summer | LE | 26.83 (± 7.92)[a] | 22.33 (± 12.06)[a] | 8.25 (± 6.30)[b] | 18.71 (± 4.46)[a] |
| | ME | 26.89 (± 9.18)[a] | 19.43 (± 13.53)[ab] | 22.13 (± 7.32)[ab] | 16.00 (± 11.72)[b] |
| | TR | 33.13 (± 18.69) | 29.00 (± 15.43) | 27.50 (± 11.21) | 16.50 (± 1.73) |
| | | **Terrestrial Percentage (%)** | | | |
| Spring | LME | 11.64 (± 13.12)[a] | 20.93 (± 19.94)[a] | 80.00 (± 34.64)[b] | — |
| | TR | 10.19 (± NA) | 16.43 (± 9.52) | 28.07 (± 15.57) | — |
| Summer | LE | 2.27 (± 1.44)[a] | 28.56 (± 26.61)[b] | 21.16 (± 21.08)[bc] | 7.65 (± 7.12)[ac] |
| | ME | 15.30 (± 7.38) | 17.81 (± 18.83) | 12.16 (± 7.49) | 5.95 (± 5.60) |
| | TR | 16.99 (± 12.69) | 27.60 (± 30.14) | 17.00 (± 7.56) | 14.91 (± 9.10) |
| | | **Energy Density (J/m$^3$)** | | | |
| Spring | LME | 99.02 (± 128.88)[a] | 6.72 (± 6.99)[b] | 3.00 (± 5.23)[b] | — |
| | TR | 19.21 (± NA) | 24.61 (± 23.97) | 12.32 (± 11.41) | — |
| Summer | LE | 11.09 (± 7.27)[a] | 6.14 (± 6.09)[a] | 1.82 (± 2.09)[b] | 10.46 (± 10.80)[a] |
| | ME | 10.16 (± 9.86)[ab] | 5.24 (± 4.26)[a] | 27.07 (± 17.88)[b] | 18.94 (± 32.05)[ab] |
| | TR | 10.74 (± 10.56) | 32.43 (± 54.57) | 8.00 (± 8.62) | 12.73 (± 12.90) |

LME = combined LE and ME sites for spring samples only. Different letters indicate significant ($P < 0.05$) pairwise year differences by section, based on Tukey's HSD.

the same taxa as observed for LE (S2B Fig), but overall composition did not change to the same degree (Fig 4B). The majority of Ephemeroptera and Plecoptera families in ME decreased in density during dam removal but did not disappear altogether as in LE. Similar to LE, Dipterans became numerically abundant in ME during dam removal (Fig 4B). By 2013, there was a net loss of eight aquatic and 11 terrestrial families in ME compared to pre-dam removal. Unlike LE, more families continued disappearing in 2014. Trichoptera was reduced from five families to one by 2014.

Examined all together, the largest relative change in invertebrate composition occurred in the spring in LME (Fig 6). TR spring multivariate centroids shifted much less from year to year, and moved largely in a perpendicular direction from LME. In summer, the trajectory of change in LE largely paralleled that of LME through 2013, but then reversed in 2014. ME centroids moved much less over time, and either perpendicular or opposite to LE. By 2014, LE and ME invertebrate composition was more similar than before dam removal. TR summer centroids changed little in position from before dam removal to 2012, but then moved in parallel with LE. Unlike LE, TR centroids continued to move further away from pre dam removal composition in 2014.

When we converted numeric drift density into energetic density, the same general patterns held over time (Fig 5)—but decreases in invertebrate energy availability during dam removal were not as severe (Table 4). For instance, numeric density decreased by 85% in LE in 2012, but energetic density only decreased by 45%. Whereas numeric drift density significantly decreased during all years of dam removal in LE, only 2013 energetic density was lower than

**Table 5. River PERMANOVA results for spring and summer invertebrate drift for: (a) Main tests of year, section, and year x section effects, and (b) Pairwise comparisons for year x section interaction.**

**A**

| Season | Source | df | SS (III) | ECV | Perm | Pseudo-F | P |
|---|---|---|---|---|---|---|---|
| Spring | Year | 2 | 10198 | 21.64 | 9897 | 2.28 | **0.003** |
| | Section | 1 | 5554 | 19.92 | 9906 | 2.49 | **0.009** |
| | Year x Section | 2 | 4979 | 9.15 | 9884 | 1.11 | 0.307 |
| | Residual | 17 | 37970 | 47.26 | | | |
| Summer | Year | 3 | 22275 | 16.83 | 9846 | 3.99 | **< 0.001** |
| | Section | 3 | 17860 | 13.63 | 9844 | 3.20 | **< 0.001** |
| | Year x Section | 9 | 29749 | 15.9 | 9786 | 1.78 | **< 0.001** |
| | Residual | 82 | 152670 | 43.15 | | | |

**B**

| Pairs | | df | Perm. | t | P | df | Perm. | t | P | df | Perm. | t | P |
|---|---|---|---|---|---|---|---|---|---|---|---|---|---|
| Spring | | | | LME | | | | | | | | TR | |
| Pre | 2012 | 8 | 210 | 1.23 | 0.067 | | | | | 3 | 5 | 0.96 | 0.557 |
| Pre | 2013 | 8 | 210 | 2.52 | **0.004** | | | | | 3 | 5 | 0.92 | 0.504 |
| 2012 | 2013 | 6 | 35 | 1.54 | 0.085 | | | | | 6 | 35 | 1.03 | 0.386 |
| Summer | | | | LE | | | | ME | | | | TR | |
| Pre | 2012 | 18 | 9626 | 2.27 | **< 0.001** | 13 | 4325 | 1.09 | 0.255 | 9 | 330 | 0.85 | 0.790 |
| Pre | 2013 | 17 | 9312 | 3.05 | **< 0.001** | 15 | 8072 | 1.93 | **< 0.001** | 9 | 330 | 1.16 | 0.161 |
| Pre | 2014 | 16 | 8518 | 2.37 | **< 0.001** | 14 | 6620 | 1.73 | **< 0.001** | 9 | 330 | 1.37 | 0.017 |
| 2012 | 2013 | 15 | 8124 | 1.84 | **0.002** | 12 | 2873 | 1.44 | **0.008** | 6 | 35 | 1.06 | 0.349 |
| 2012 | 2014 | 14 | 6614 | 1.42 | 0.017 | 11 | 1707 | 1.37 | 0.027 | 6 | 35 | 1.31 | 0.160 |
| 2013 | 2014 | 13 | 5060 | 1.82 | **0.003** | 13 | 5050 | 1.27 | 0.084 | 6 | 35 | 0.91 | 0.534 |

Test based on partial sums of squares (SS) and permutation of residuals under a reduced model, Perm = total number of unique possible permutations, df = denominator degrees of freedom, and P-values are based on permutations. ECV = square root of the estimated components of variation in the model. Pseudo-F and pseudo-t values are the multivariate analog to the univariate F and t statistic. Monte carlo P-values are reported for tests with < 40 possible permutations. Taxonomic resolution is at the family level for square-root transformed invertebrate numerical density. Bolded black values indicate $P \leq 0.05$ for main test and spring pairwise comparisons, and $\leq 0.0083$ (Bonferroni adjusted) for summer pairwise comparisons.

**Table 6. Univariate diet metrics (means ± 1 SD) for coho and Chinook juvenile salmon collected in the estuary.**

| Season | Pre | 2013 | 2014 |
|---|---|---|---|
| | **Number of empty stomachs** | | |
| Spring | 4 | 16 | 2 |
| Summer | 0 | 7 | 0 |
| | **Number of prey items (Total individuals)** | | |
| Spring | 24.8 (± 23.8)[ab] | 16.0 (± 57.2)[a] | 45.2 (± 81.0)[b] |
| Summer | 21.8 (± 18.2)[a] | 21.5 (± 26.7)[a] | 34.3 (± 37.4)[b] |
| | **Taxa richness (Number of unique prey taxa)** | | |
| Spring | 3.7 (± 2.6)[a] | 2.4 (± 2.2)[b] | 6.0 (± 3.1)[c] |
| Summer | 4.5 (± 2.7) | 4.4 (± 3.1) | 5.1 (± 2.5) |
| | **Terrestrial taxa (percentage of total KJ)** | | |
| Spring | 6.0 (± 17.1)[a] | 24.1 (± 37.4)[ab] | 15.0 (± 22.9)[b] |
| Summer | 38.7 (± 39.6) | 28.0 (± 35.1) | 25.4 (± 31.8) |
| | **Prey energy (Total KJ)** | | |
| Spring | 0.27 (± 0.35) | 0.24 (± 1.05) | 0.53 (± 1.00) |
| Summer | 0.30 (± 0.32) | 0.28 (± 0.35) | 0.25 (± 0.22) |

Data are calculated on a per stomach basis. Different letters indicate significant ($P < 0.05$) pairwise year differences by section, based on 1-way ANOVA and Tukey's HSD.

before removal ($P < 0.05$). In ME, drift energy was actually greater in 2013 than in 2012 ($P < 0.05$). Differences between total numeric and energetic density were largely due to the higher energetic content of terrestrial taxa. Over half of drift energy in ME 2013 was supplied by terrestrial taxa, as was also the case for LE in 2012 (Fig 5).

## Estuary fish diet composition

The number of fish used for diet analyses varied by season and year, ranging from 53 to 88 individuals (Table 1). Estuarine Chinook and coho selected for lavage sampling were between 45 to 195 mm in length (mean = 96.7 mm, SD = 22.1 mm) and mass ranged from 0.9 g to 55 g (mean = 8.5 g, SD = 6.5 g). As less than 10% of captured fish were YOY, we combined size classes for analyses. Fish diet was not significantly different between Chinook and coho (ANOSIM R = 0.05, $P = 0.08$), so we analyzed both species together. The relative abundance of the two species varied over the study period. Chinook comprised 32% of diet samples before dam removal versus 30% during dam removal in spring, and 7% before versus 48% during dam removal in summer. We observed a nearly eight-fold increase in the number of empty stomachs in 2013 compared to before removal (Table 6).

The mean number of prey items per stomach and unique prey items per fish stomach were lowest in spring 2013 and highest in spring 2014 (Table 6; Tukey HSD, $P < 0.05$). We did not detect a significant difference in mean prey energy over time in either season ($P > 0.05$) (Figs 7A and 8A). However, the proportion of aquatic and terrestrial invertebrates in fish diets changed seasonally and across years. The proportion of terrestrial and fish prey items increased substantially in spring 2013 compared to before dam removal (1-way ANOVA, $P < 0.05$; Table 6), while the proportion of aquatic species declined (Fig 7B). The proportion of terrestrial prey items in summer was not significantly different between years ($P > 0.05$), but the proportion of fish prey increased during removal, particularly in 2013 (Fig 8B).

The composition of taxa contributing to fish diet energy varied significantly by year for both spring and summer (PERMANOVA, spring $P < 0.001$, summer $P < 0.004$; Table 7A, Figs 7C and 8C). Pairwise tests between years showed that diet composition before dam removal was significantly different than both years (2013-2014) of dam removal; there was no significant difference between 2013 and 2014 (Table 7B). Diet composition nearly converged in 2013 in spring and summer (Fig 9) before diverging in 2014.

The loss of brackish-water taxa (e.g., the Amphipoda family Corophiidae) from the estuary during dam removal strongly contributed to the difference between diet composition before and during dam removal (S3 Fig). New items were also found in fish stomachs during dam removal, including small fish, plankton (e.g., Cyclopoida), and freshwater gastropods. There was also an increased reliance on aquatic-origin taxa with worm-like larval stages (e.g., Chironomini, Orthocladiinae, Tanypodinae) during dam removal.

## River fish diet composition

We collected diet data from 764 trout across all seasons, river sections, and years. Fish size distribution was very similar before (n= 284) and during (n = 480) dam removal, with median FL 82–83 mm in both time periods. FL did not vary by section in spring, but in summer FL in TR (median = 89 mm) was significantly longer than in LE (median = 77 mm) (Tukey HSD, $P < 0.05$). Across all study sections, a third of all captured fish were YOY (FL $<= 70$ mm) and the remainder were 1+ age class (71–200 mm). We observed similar size distributions between spring (n = 237, median = 84 mm) and summer (n = 527, median = 82). There was no change in the proportion of empty stomachs (mean = 5.2%) over time in any section.

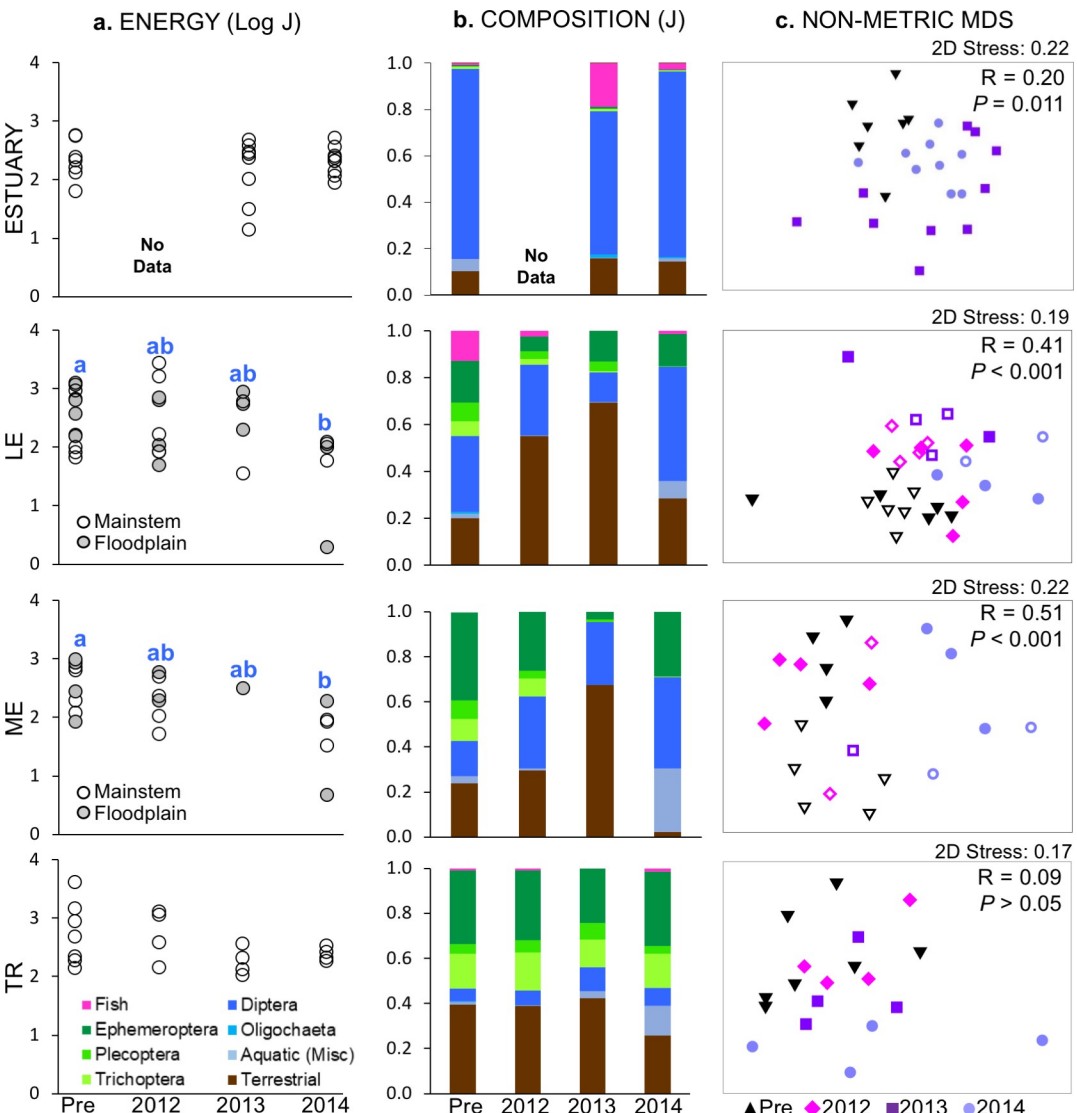

**Fig 8. Summer fish diet by section and sample year.** Data are plotted as (a) mean total prey energy per fish by site, (b) relative composition by major taxonomic groups (section means), and (c) nMDS, with global sample statistic and significance for one-way ANOSIM by year; note that ordinations with 2D stress values > 0.2 may not adequately represent data structure [36]. All data are in J. For panel a, different letters indicate significant year differences ($P < 0.05$) within a section. For panel c, floodplain sites are indicated with outlined symbols.

We detected minor differences in diet metrics between YOY and 1+ fish. As expected, there was significantly more prey energy per fish in 1+ fish than YOY in both spring and summer (3-way ANOVA, size class, $P < 0.001$). In summer, the difference in prey energy between size classes was greater before dam removal and in 2012 relative to 2013 or 2014 (size x year, $P < 0.05$). There was no significant effect of size class on the mean number of prey items, nor the proportion of diet comprised of terrestrial taxa ($P > 0.05$). In terms of overall diet composition, we did not detect significant differences between YOY and 1+ in either season (ANOSIM, $P < 0.05$, spring $R = 0.11$, summer $R = 0.06$). We therefore grouped all trout sizes together for subsequent diet analyses.

**Table 7. Estuary prey energy PERMANOVA results for: (a) Main test of year, and b) Pairwise comparisons for year.**

**A**

| Season | Source | df | SS (III) | ECV | Perm | Pseudo-*F* | *P* |
|---|---|---|---|---|---|---|---|
| Spring | Year | 2 | 13263 | 28.056 | 998 | 2.6 | **0.001** |
| | Residual | 13 | 33129 | 50.482 | | | |
| Summer | Year | 2 | 7672 | 21.135 | 990 | 2.01 | **0.004** |
| | Residual | 10 | 19121 | 43.727 | | | |

**B**

| Season | Pairs | | df | | Perm. | t | *P* |
|---|---|---|---|---|---|---|---|
| Spring | Pre | 2013 | 10 | | 566 | 1.58 | **0.003** |
| | Pre | 2014 | 9 | | 314 | 1.94 | **0.003** |
| | 2013 | 2014 | 7 | | 126 | 1.16 | 0.155 |
| Summer | Pre | 2013 | 7 | | 126 | 1.53 | **0.048** |
| | Pre | 2014 | 6 | | 35 | 1.5 | **0.023** |
| | 2013 | 2014 | 7 | | 125 | 1.21 | 0.062 |

Test based on partial sums of squares (SS) and permutation of residuals under a reduced model. Perm = total number of unique possible permutations, df = denominator degrees of freedom, and *P*-values are based on permutations. ECV = square root of the estimated components of variation in the model. Pseudo-F and pseudo-t values are the multivariate analog to the univariate F and t statistic. Taxonomic resolution is at the family level for fourth-root transformed invertebrate counts and energy. Bolded black values indicate *P* < 0.05.

Across all diet samples, we identified 245 unique prey taxa: 148 of aquatic origin and 97 from the terrestrial zone. Most prey taxa were insects, comprising 76% of the 166 families present in fish stomachs. Mites and spiders represented another 7% of families, with the remaining

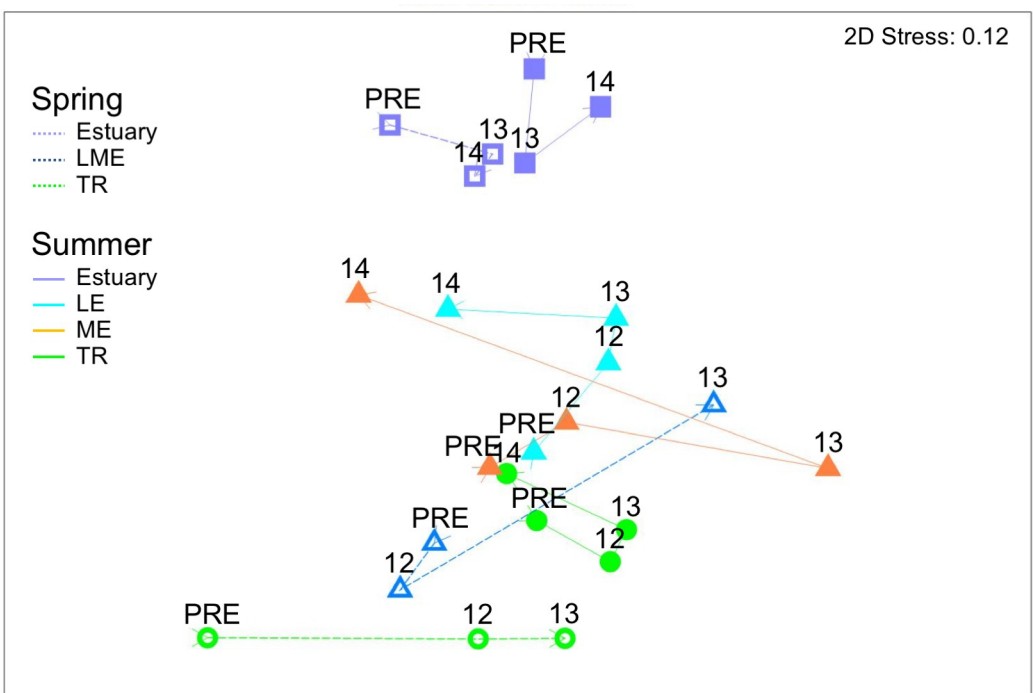

**Fig 9. Fish diet nMDS plots of data centroids.** Trajectories show direction of change by season, year and section for energetic composition. All data are fourth-root transformed.

**Table 8. Univariate diet metrics (means ± 1 SD) for trout collected in the river.**

| Season | Section | PRE | 2012 | 2013 | 2014 |
|---|---|---|---|---|---|
| | | **Prey energy (Total KJ)** | | | |
| Spring | LME | 0.24 (± 0.17) | 0.24 (± 0.33) | 0.26 (± 0.28) | — |
| | TR | 0.06 (± NA) | 0.83 (± 0.55) | 0.67 (± 0.42) | — |
| Summer | LE | 0.51 (± 0.45)[a] | 0.69 (± 0.93)[ab] | 0.45 (± 0.34)[ab] | 0.08 (± 0.05)[b] |
| | ME | 0.53 (± 0.36)[a] | 0.28 (± 0.21)[ab] | 0.31 (± NA)[ab] | 0.08 (± 0.07)[b] |
| | TR | 1.07 (± 1.42) | 0.73 (± 0.56) | 0.20 (± 0.12) | 0.25 (± 0.07) |
| | | **Number of prey items (Total individuals)** | | | |
| Spring | LME | 9.88 (± 7.18) | 7.40 (± 10.57) | 3.02 (± 2.70) | — |
| | TR | 5.67 (± NA) | 8.75 (± 3.28) | 12.13 (± 4.56) | — |
| Summer | LE | 21.70 (± 15.09) | 12.14 (± 6.58) | 13.30 (± 14.04) | 20.85 (± 19.07) |
| | ME | 17.25 (± 15.16) | 23.96 (± 13.08) | 18.14 (± NA) | 13.90 (± 10.39) |
| | TR | 22.40 (± 12.12) | 21.42 (± 4.51) | 14.15 (± 8.72) | 16.23 (± 10.24) |
| | | **Taxa richness (Number of unique prey taxa)** | | | |
| Spring | LME | 3.61 (± 1.37) | 2.72 (± 2.25) | 2.15 (± 1.94) | — |
| | TR | 3.11 (± NA) | 5.24 (± 1.89) | 6.48 (± 1.68) | — |
| Summer | LE | 4.73 (± 1.85) | 5.35 (± 2.89) | 7.30 (± 5.24) | 3.33 (± 2.17) |
| | ME | 4.55 (± 1.79) | 4.77 (± 2.09) | 6.14 (± NA) | 3.35 (± 1.55) |
| | TR | 8.14 (± 3.35) | 7.85 (± 1.47) | 6.78 (± 1.70) | 5.17 (± 2.85) |
| | | **Terrestrial taxa (Percentage of Total KJ)** | | | |
| Spring | LME | 8.12 (± 7.84)[a] | 19.20 (± 16.96)[ab] | 66.28 (± 0.55)[b] | — |
| | TR | 5.21 (± NA) | 22.09 (± 18.43) | 17.01 (± 14.33) | — |
| Summer | LE | 18.67 (± 17.67)[a] | 53.75 (± 28.28)[b] | 69.34 (± 30.75)[b] | 28.33 (± 26.14)[ab] |
| | ME | 23.75 (± 29.42)[a] | 27.61 (± 28.17)[a] | 67.31 (± NA) | 2.36 (± 4.24)[b] |
| | TR | 39.05 (± 21.69) | 38.77 (± 21.81) | 42.34 (± 20.27) | 25.81 (± 17.40) |

Data are calculated on a per stomach basis. Different letters indicate significant ($P < 0.05$) pairwise year differences by section, based on Tukey's HSD.

32 families a mix of collembola, crustaceans, gastropods, worms, centipedes, and millipedes. We observed limited piscivory across freshwater sample sites. Of the 764 diet samples, only 11 contained identifiable fish remains.

Mean prey energy declined significantly in sediment-impacted section during the summer, but not in the spring (Table 8). There were no significant year-to-year changes in prey energy in TR for either season. In both LE and ME, mean prey energy per fish was greater before dam removal than in 2014 (Tukey HSD, $P < 0.05$) (Fig 8A). We did not detect an interaction effect of year x section on prey energy ($P > 0.05$). Nor did we detect significant yearly differences in any season or section on either mean number or unique types of prey per stomach. Habitat Type (mainstem versus floodplain) was not a significant factor on any of the above three metrics (year x habitat, $P > 0.05$).

The proportion of total prey energy provided by terrestrial taxa changed during dam removal in sediment-impacted sections, but not in TR. In both seasons, we detected a significant year effect (2-way ANOVA, $P < 0.05$), but not an interaction of year x season. In LME spring fish diets, significantly more energy came from terrestrial taxa in 2013 (mean = 66%) than before (8%) dam removal (Tukey HSD $P < 0.05$) (Fig 7B). The same pattern was observed in summer for LE, with terrestrial percentage higher in 2012 and 2013 than before dam removal ($P < 0.05$) (Fig 8B). In ME, terrestrial contribution nearly tripled in 2013, but then was significantly lower in 2014 than all other years ($P < 0.05$). We also observed a significantly

**Table 9. River PERMANOVA results for spring and summer fish diet composition for: (a) Main tests of year, section, and year x section effects, and (b) Pairwise comparisons for year x section interaction.**

**A**

| Season | Source | df | SS (III) | ECV | Perm | Pseudo-F | P |
|---|---|---|---|---|---|---|---|
| Spring | Year | 2 | 9471 | 18.05 | 9879 | 1.66 | **0.005** |
| | Section | 1 | 5086 | 16.71 | 9912 | 1.78 | **0.017** |
| | Year x Section | 2 | 4879 | -11.9 | 9862 | 0.86 | 0.747 |
| | Residual | 16 | 45625 | 53.4 | | | |
| Summer | Year | 3 | 31342 | 22.34 | 9825 | 3.61 | **< 0.001** |
| | Section | 3 | 18362 | 14.35 | 9834 | 2.11 | **< 0.001** |
| | Year x Section | 9 | 27907 | 6.64 | 9684 | 1.07 | 0.201 |
| | Residual | 66 | 191290 | 53.84 | | | |

**B**

| Pairs | | df | Perm. | t | P | df | Perm. | t | P | df | Perm. | t | P |
|---|---|---|---|---|---|---|---|---|---|---|---|---|---|
| Spring | | | LME | | | | | | | | TR | | |
| Pre | 2012 | 8 | 210 | 1.3 | **0.041** | | | | | 3 | 5 | 0.93 | 0.507 |
| Pre | 2013 | 7 | 84 | 1.51 | **0.012** | | | | | 3 | 5 | 1.13 | 0.335 |
| 2012 | 2013 | 5 | 35 | 1.14 | 0.298 | | | | | 6 | 35 | 1.01 | 0.421 |
| Summer | | | LE | | | | ME | | | | TR | | |
| Pre | 2012 | 18 | 9605 | 1.74 | **< 0.001** | 13 | 4315 | 1.45 | **0.008** | 9 | 330 | 1 | 0.448 |
| Pre | 2013 | 14 | 3923 | 1.6 | **< 0.001** | 8 | 10 | 1.1 | 0.304 | 9 | 330 | 1.02 | 0.387 |
| Pre | 2014 | 14 | 3921 | 1.68 | **< 0.001** | 12 | 1985 | 1.69 | **< 0.001** | 9 | 330 | 1.28 | 0.035 |
| 2012 | 2013 | 12 | 1981 | 1.11 | 0.176 | 5 | 7 | 0.91 | 0.524 | 6 | 35 | 0.92 | 0.539 |
| 2012 | 2014 | 12 | 1988 | 1.64 | **0.001** | 9 | 462 | 1.7 | **0.002** | 6 | 35 | 1.18 | 0.244 |
| 2013 | 2014 | 8 | 126 | 1.29 | **0.03** | 4 | 6 | 1.15 | 0.293 | 6 | 35 | 1.08 | 0.415 |

Test based on partial sums of squares (SS) and permutation of residuals under a reduced model. Perm = total number of unique possible permutations, df = denominator degrees of freedom, and *P*-values are based on permutations. ECV = square root of the estimated components of variation in the model. Pseudo-F and pseudo-t values are the multivariate analog to the univariate F and t statistics. Monte carlo *P*-values are reported for tests with < 40 possible permutations. Taxonomic resolution is at the species level for fourth-root transformed energetic density. Bolded black values indicate $P \leq 0.05$ for main test and spring pairwise comparisons, and $\leq 0.0083$ (Bonferroni adjusted) for summer pairwise comparisons.

higher proportion of terrestrial taxa in floodplain channels than in mainstem (2-way ANOVA, Habitat Type effect, P < 0.001), but no interaction effect with year.

Overall taxonomic composition of fish diets changed during dam removal in both spring and summer (PERMANOVA two-way crossed design; *P* < 0.05) (Table 9A) (Figs 7C and 8C). Spring diet in LME was significantly different in 2012 and 2013 relative to before removal (Table 9B). In summer, diet composition in LE was significantly different in every pair-wise comparison except 2012/2013 and 2013/2014. Diet differed between all years in ME except 2013, when low fish capture limited statistical power. We also observed an effect of Habitat Type on diet composition (PERMANOVA; *P* < 0.05), but no interaction of habitat type and year.

Before dam removal, EPT taxa (from the insect orders Ephemeroptera, Plecoptera, Trichoptera) were the dominant source of aquatic energy in fish diets across all study sections, contributing 80–93% of total joules in spring (Fig 7B) and 32–86% in summer (Fig 8B). By 2013, only 15% of diet energy in LME derived from EPT taxa. Instead, the majority of energy came from terrestrial taxa and Oligochaetes. Taxa that contributed the most to differences in LME fish diets between years were all aquatic save one (S4A Fig). Declines in EPT taxa consumption over time contributed the most to dissimilarities between years.

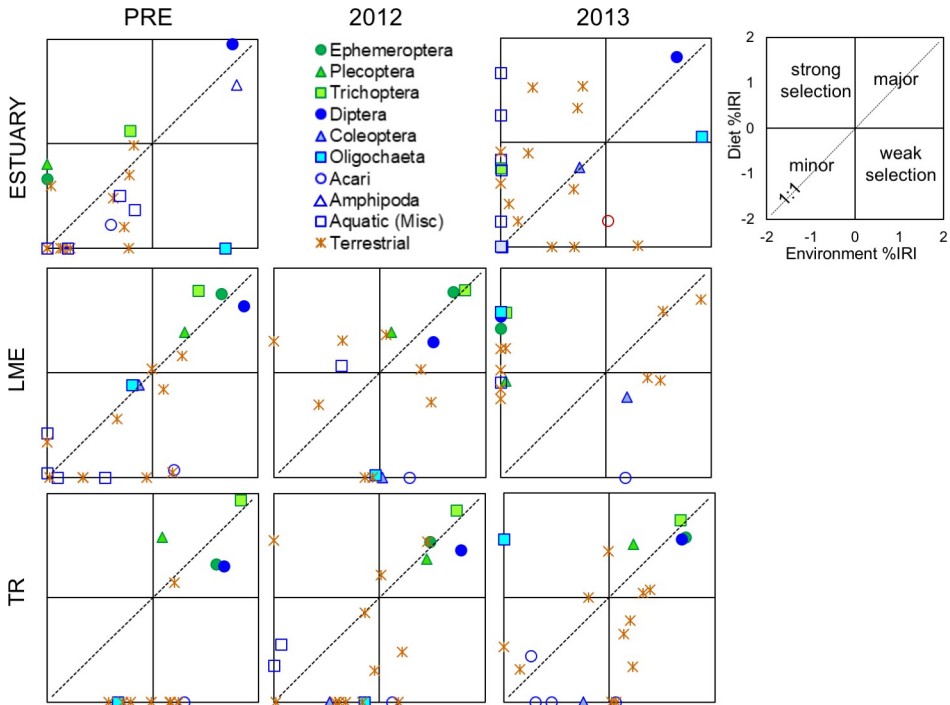

**Fig 10. Index of Relative Importance (% IRI) for invertebrates present in spring environmental (x-axis) and diet (y-axis) samples.** Data are $\log_{10}$ transformed (i.e., 0.1% = - 2, 100% = + 2), and plotted by study section and year. Each terrestrial data point represents a different order. Orders that contributed < 0.1% to IRI are plotted at -2. Graphs do not include taxa consumed by fish but not present in environmental samples (i.e., plankton in the estuary, and fish in estuary and river).

In the summer, we observed widespread taxonomic changes in LE and ME summer diets across years (S4B Fig). In LE, 33 aquatic families decreased in abundance in 2012 while 22 terrestrial families increased—with a net loss of 17 aquatics and gain of three terrestrials. In 2013, there was a net loss of 30 aquatic and 14 terrestrial families relative to before dam removal. During the same time period in ME, there was a net loss of 20 aquatic and terrestrials 26 taxa. Unlike LE, the majority of families in ME continued to decrease in abundance from 2013 to 2014. By summer 2014, no Trichoptera or Plecoptera taxa were found in the diets of fish from LE or ME (Fig 8B).

## Relationship between fish diet and prey availability in the estuary

We did not detect any change over time in the extent of overlap between invertebrate availability and fish diet based on Bray-Curtis similarity coefficients calculated at each site (1-way ANOVA, $P > 0.05$). There was a high degree of within-season variability. In the spring, coefficient values ranged from 3.65–32.93 (mean = 17.98) before dam removal and from 5.51–26.53 (mean = 13.40) during. In the summer, values ranged from 0.72–13.45 (mean = 4.14) before dam removal and from 0.34–30.88 (mean = 10.32) during.

Aquatic Diptera and Amphipods were the most important diet item for estuary fish before dam removal, and were consumed in relative proportion to their availability in the environment (Figs 10 and 11). While Diptera remained an important food source over time, Amphipods disappeared altogether during dam removal. Selection of EPT taxa by fish increased during dam removal, but with the availability of these taxa at or near zero, their relative importance to fish diet was small. The relative importance of terrestrial taxa did not change greatly

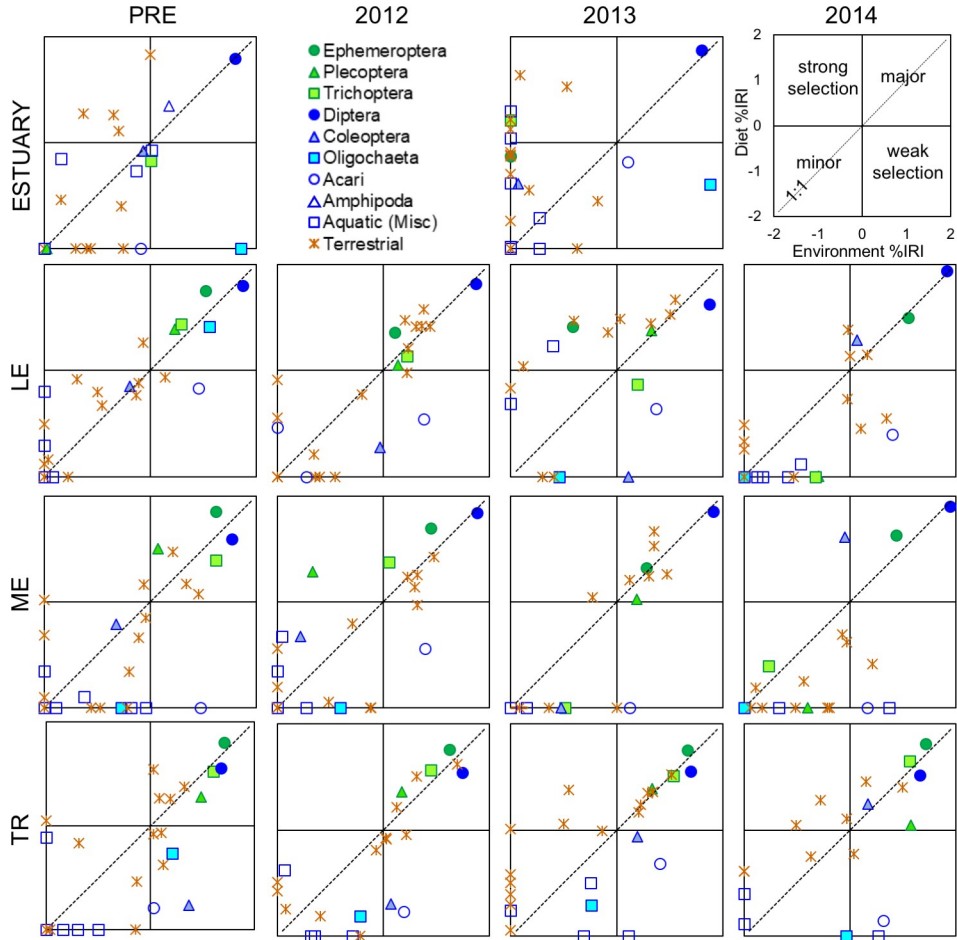

**Fig 11. Index of Relative Importance (% IRI) for invertebrates present in summer environmental (x-axis) and diet (y-axis) samples.** Data are $\log_{10}$ transformed (i.e., 0.1% = - 2, 100% = + 2), and plotted by study section and year. Each terrestrial data point represents a different order. Orders that contributed < 0.1% to IRI are plotted at -2. Graphs do not include taxa consumed by fish but not present in environmental samples (i.e., plankton in the estuary, and fish in estuary and river).

during dam removal in summer (Fig 11B), but increased in spring (Fig 10A). While the prevalence of Oligochaetes in the environment increased during dam removal, these taxa were strongly selected against by fish in all seasons and years.

## Relationship between fish diet and prey availability in the river

Based on Bray-Curtis similarity coefficients, overlap between invertebrate availability and fish diet increased significantly in LE in summer 2014 relative to all prior sample years (Tukey HSD, $P < 0.05$). We did not detect changes in similarity over time in spring, nor in summer in ME or TR. Prior to dam removal, mean Bray-Curtis similarity in spring was 13.89 in LME and 34.96 in TR. These values increased over time in LME and decreased in TR, but were highly variable. In summer, mean Bray-Curtis similarity across all sections prior to dam removal ranged from 11.29 to 16.81. While both LE and ME trended towards higher similarity during dam removal, these changes were only significant in LE for 2014, when Bray-Curtis similarity across all sample sites reached a mean of 33.66.

Greater similarity between invertebrate availability and fish diet in 2014 may reflect the increased relative importance of a smaller subset of taxa—particularly aquatic Dipterans (Fig 11). Aquatic Dipterans were among the largest contributors to drift IRI, and were typically consumed in the same relative proportion to their availability or slightly selected against. As EPT taxa disappeared from sediment-impacted sections during dam removal, aquatic Dipterans became increasingly important. By summer 2014, Diptera was by far the largest contributor to both drift and diet IRI in LE and ME. In reference section TR, the importance of EPT to fish remained consistent across years, as did their availability in the environment.

In the first two years of dam removal, terrestrial taxa that were previously selected against or were relatively minor contributors to both diet and drift became some of the most important prey sources for fish in LME, LE, and ME. Lepidoptera and Hemiptera taxa were particularly large contributors. The relative importance of terrestrial taxa to fish in TR was fairly consistent across time. As availability of EPT taxa decreased during dam removal in impacted sections, selectivity of these taxa by fish increased. The water mites Acari, while relatively plentiful throughout dam removal, were strongly selected against in all sections and years.

## Discussion

The massive release of reservoir sediments during Elwha dam removal significantly changed downstream invertebrate communities, but had a much smaller effect on the energetic content of juvenile salmonid diet. Food web effects were similar across freshwater and estuary habitats, while we did not detect any significant changes in non-sediment impacted tributaries. Invertebrate composition changed in all sections impacted by dam removal, but density declines were greatest downstream of the Elwha Dam. The effect this had on fish diet was buffered in part by concurrent shifts in invertebrate taxonomic composition and feeding selectivity.

The amount of energy in fish diets remained relatively stable through dam removal, but the sources of this energy changed. In the estuary, these shifts were likely driven by a hydrological transition from a brackish to a freshwater environment following persistent sediment deposition at the river mouth [37]. In the river, energy provided to fish by EPT taxa before dam removal was replaced by Chironomids and other more sediment-tolerant taxa. In both the estuary and river, terrestrial taxa comprised a greater portion of fish diet during than before dam removal.

Changes in prey availability and diet composition varied in relation to the timing and magnitude of sediment released during dam removal. Food web shifts were first observed downstream of the Elwha Dam following WY 2012 sediment transport from Lake Aldwell, and between the dams following WY 2013 transport from Lake Mills. Across all sections, prey availability was at its lowest in 2013 during the period of maximum sediment transport [5,19]. Invertebrate density declines and associated changes in fish diet were magnified in the spring relative to summer.

We explore in more detail below the processes and factors responsible for these outcomes. In addition to the downstream dispersal of reservoir sediments, we examine the roles anadromous fish recolonization and other natural sources of variability played in our results. We conclude with a discussion of the role of ecosystem resilience during dam removal, and what factors contribute to intrinsic resiliency.

### Prey availability changed the most downstream of the Elwha Dam

During summer, the estuary and LE were the only portions of the study area where we detected significant density changes in prey during dam removal. Invertebrate density declined by > 50% in the estuary and > 95% in LE. These sections not only experienced

sediment impacts from removal of both dams, but were also disturbed over a longer time period (because the timing of Glines Canyon Dam removal delayed the largest sediment impacts to ME; Fig 2E). Due to a lower gradient, more fine sediment deposition also occurred downstream of the Elwha Dam than it did between the two dams (Fig 2D and 2E).

Prey composition significantly changed across all sediment-impacted sections, but was most pronounced downstream of the Elwha Dam. In the estuary, the shift towards freshwater habitat resulted in the loss of brackish invertebrate species [37]. High levels of fine sediment deposition also favored worms and other burrowing invertebrates (e.g., Chironomidae) [26]. The relative abundance of Chironomidae also increased across river study sections, while the proportion of EPT taxa decreased. By 2014, no Plecoptera or Trichoptera taxa were observed in LE drift samples, and Chironomidae comprised > 90% of the drift.

Across both river and estuarine habitats, the proportion of terrestrial prey taxa increased during dam removal. This was due to decreased aquatic densities, but the absolute density of terrestrial invertebrates also increased. In LE, terrestrial density more than doubled in 2012, but then largely disappeared in 2013. A similar pattern was observed in ME in 2013 and 2014. Terrestrial invertebrates may have benefitted from the early stages of sediment disturbance, potentially responding to the initial flush of nutrients and organic matter released from the reservoir sediment deposits and dispersed across downstream floodplain and riparian habitats [38,39].

Terrestrial invertebrates are a valuable food resource for juvenile salmonids [40,41]. Others have noted shifts in juvenile salmon foraging towards these allochthonous prey sources when they become seasonally available [42,43]. These taxa tend to be larger-bodied than many aquatic invertebrates, and their lower water content translates into higher energy content [44]. In addition to high caloric value, terrestrial taxa also contributed a large portion of prey diversity at our study sections. Terrestrials floating on the surface of the water column may also have been easier for fish to catch than aquatic taxa in the benthos or drift during periods of high turbidity [22].

## Total fish diet energy did not change greatly, but energy sources did

The average amount of energy in fish diet did not significantly change during dam removal in the estuary, and only during 2014 in the river. However, the sources of energy did change—consistent with the compositional shifts in fish diet we observed. Despite the huge decrease in invertebrate numeric density during the peak of dam removal sediment effects (2012–2013), fish maintained a level of energy in their diets similar to before dam removal. However, when river drift densities began to rebound in 2014, fish diet energy was actually lower than before dam removal. These patterns are due to differences in the energetic content of prey throughout the three years of dam removal.

Higher energetic content of prey in the first two years of dam removal was due to the increased prevalence of terrestrial-origin taxa in the river and to increased piscivory in the estuary. In LE, terrestrial taxa comprised 21-29% of individuals in the drift over 2012–2013, but contributed over half of the energy content. In the estuary, piscivory contributed 50–60% of total diet energy in the spring of 2013 and 2014, while the contribution of other aquatic food sources dropped to nearly zero (Fig 9). In 2014, the lower energy content in fish diet likely reflected the higher proportion of small-bodied taxa such as Chironomids (Diptera) in the diet (Fig 8B).

While fish generally managed to maintain similar energy levels in their diets during dam removal, they may have had to expend more energy to obtain food. Based on the high turbidity levels measured in the river and estuary (especially in the spring), it is likely that visual foraging

was compromised [21,23,45]. In the estuary, the number of empty stomachs was six times higher in 2013 than any other sample year, a time period that corresponded to the lowest observed prey densities and the highest turbidity.

We did not detect more empty stomachs in the river, but did observe changes in fish densities during dam removal. Based both on our own electrofishing catch rates (unpublished) and smolt outmigration data collected by the Lower Elwha Klallam Tribe [46], there was a sharp decline in juvenile salmonid densities in the river between 2012 and 2014. This decline was likely due to a combination of direct (e.g., decreased egg-to-fry survival) and indirect effects (e.g., low benthic invertebrate production). Decreased competition for remaining food resources may be another reason we did not detect larger differences in diet energy content during dam removal.

### Timing and mechanisms of change

Downstream of the Elwha dams, the primary drivers of aquatic food web change were largely caused by sediment released from the former reservoirs. Elevated turbidity and suspended sediment concentrations can affect both invertebrates and fish directly by causing gill abrasion and erosion of mucus coatings, decreasing feeding efficiency, increasing stress levels, and reducing overall habitat quality [46–48] and references therein. Indirect effects include reduced primary production from decreased light availability [49]. In experimental studies, increased turbidity leads to decreased growth and increased emigration of juvenile salmonids [22,50,51].

In addition to elevated turbidity, invertebrates and fish were also impacted by high levels of fine sediment deposition during dam removal [5,52,53]. In 2012, approximately 7% of transported reservoir sediment was deposited in LE, filling pools, slow-velocity channel margins, floodplain channels, and cobble interstices with up to 0.5 m of primarily sand and mud [52,54] (Fig 2D). Another 1% of fine-grained sediment was deposited in the estuary in 2012 [55], resulting in up to 1 m of sediment deposition in some areas [26].

When the sediment pulse peaked in winter 2013, geomorphic changes increased downstream of the Elwha Dam, and were also observed between the two dams in ME. In LE, sediment deposits of up to 2.0 m were recorded, and the bed continued to become finer-grained over the spring and summer of 2013 [52]. ME experienced a 16-fold decrease in grain-size distribution and deposits up to 1.5 m [52] (Fig 2E). However, this was a transient effect. In spring 2013, the river incised through these fine deposits and by summer ME returned closer to pre-removal elevation and sediment size distributions.

During the third year of dam removal, 0.25 Mt of reservoir sediment was deposited in LE and 0.29 Mt in ME over the fall and winter of WY 2014 [5]. In the estuary, sediment deposition in excess of 1 m extended the river delta approximately 400 m seaward into the Strait of Juan de Fuca, tripling the amount of intertidal habitat [26]. In LE, the newly-deposited sediment remained in place through summer 2014, and continued to fine [56]. In ME, sediments deposited in winter were once again largely eroded away during spring flows (Fig 2E).

Repeated deposition and scouring of the river bottom creates an inhospitable environment for most benthic invertebrate taxa. Based on 26 published studies of in-stream invertebrate response to dam removal, one third showed a decrease in downstream invertebrate density after dam removal [57]. Repeated burial and erosion of bottom sediments decreases overall habitat availability for all but the most rapidly-reproducing invertebrates that can complete their life cycle between disturbance events, or those like oligochaete worms which are adapted to survival in substrates with high levels of fine sediment [48,58,59].

High levels of sediment deposition can also reduce overall habitat quality by filling interstitial spaces and smothering rocky surfaces. Interstitial habitats are important refugia that many

invertebrates use to escape from high velocity flows and predators. When these pore spaces are filled with fine sediments, density and diversity of invertebrate communities decrease [49,59–61]. Taxa that live upon the surface of substrates to feed are also vulnerable when fine substrates scour or cover sediments [48].

A less plentiful and diverse food source negatively impacts resident fish, as does the smothering of spawning gravels under a heavy layer of fine sediment [47,62]. Increased fine sediment loads have been shown to decrease growth and survival of juvenile *O. mykiss* when food webs shift towards burrowing invertebrate taxa [63]. High levels of fine sediment deposition can also reduce dissolved oxygen levels, increasing physiological stress to fish, and in extreme cases lead directly to fish mortality via hypoxia [47].

The changes we observed in the Elwha food web during dam removal are consistent with the magnitude and timing of the sediment-related changes in physical habitat described above. We observed a near complete loss of aquatic-origin invertebrates in estuary fallout traps in the spring of 2013 and a 99.9% reduction in spring 2013 river drift density following the peak sediment pulses of winter 2013. In summer 2013, drift composition in LE shifted even further from pre-dam removal, and we began to also observe taxonomic changes in ME from the release of Lake Mills sediment.

As seen in other river systems experiencing high levels of sediment deposition, Chironomidae (and particularly the subfamily Orthocladiinae) became increasingly dominant in the estuary and river over the three years of dam removal [64,65]. With their small body size, short life cycle, and fine-sediment tolerance, these midges were better able to tolerate the physical effects of dam disturbance than more sensitive orders such as Trichoptera (many of which are net spinning, or require specific substrates for case building) and Plecoptera (which frequently inhabit interstitial spaces). Although most Ephemeroptera taxa were also greatly reduced, some (such as the multivoltine *Baetis tricaudatus*) were able to weather the sediment pulse.

Changes in juvenile salmonid diets also peaked in spring 2013 at most study sections (Fig 11). In the estuary, juvenile fish began consuming large numbers of plankton, and in the estuary and river a higher proportion of terrestrial taxa were consumed in 2013. This shift to a planktivorous and surface feeding strategy is consistent with increased turbidity in the water column [45,66]. However, extreme turbidity levels—such as those seen during the winter and early spring of 2013—were likely high enough to affect all foraging strategies [67].

Declines in invertebrate density and diversity and associated changes in fish diet were magnified in the spring relative to summer (Figs 6 and 9). Most sediment transport during dam removal occurred during winter freshets or spring snowmelt, whereas low-flow summer periods typically experienced less severe turbidity and fine-sediment deposition [5]. The buffering effect of terrestrial invertebrate inputs was also not present to the same degree in spring as in summer, when terrestrial invertebrate biomass is highest [40]. Thus, spring was likely a more critically food-limited period during dam removal for juvenile salmonids, with fewer sources of both aquatic and terrestrial prey after a period of winter starvation [68].

With the sediment pulse declining in WY 2014, there were signs of biological recovery in some sections, but continued disturbance in others. In the estuary, Diptera became more dominant again in 2014. In LE, overall taxonomic composition began to shift back towards pre-dam removal levels, although EPT taxa richness remained depressed. In 2014 summer, numerical drift density rebounded closer to pre-dam removal levels in LE. Juvenile salmonid diets also contained more aquatic taxa in 2014 than 2012 or 2013. However, diet composition continued to diverge further from pre-dam removal conditions in 2014 for ME.

## Additional sources of variability

Although sediment impacts played a dominant role in the changes we observed, other factors related to dam-removal, climate, and study design also contributed to our results. In the fall of 2011, the Lower Elwha Klallam Tribe began actively transplanting returning adult coho and steelhead salmon into tributaries upstream of the Elwha Dam (including two of our study reaches) [69]. Coupled with natural recolonization, juvenile fish densities in these tributaries subsequently increased by up to an order of magnitude during our study period [69]. Associated changes in competition and predation may have increased interannual variability in our reference dataset.

It is likely that climatic conditions were another source of variability across our study period. For example, our sampling in 2014 occurred during a prolonged period of drought, whereas 2010 and 2011 were relatively wet years with higher than average flows during late summer (Fig 2B). Associated annual fluctuation in the emergence timing of invertebrates relative to our sample periods may have contributed additional variability, and speaks to the importance of collecting multiple years of data before and after in any monitoring study.

High background variability in some of our food web metrics was compounded by low sample size in some cases. High flow events during our spring sampling period reduced site accessibility and thus sample size. Our ability to detect change relative to dam removal was more limited in spring than summer. Very low fish capture rates during dam removal also limited our statistical power in terms of examining diet changes—particularly in ME.

## Ecosystem resilience and recovery

Recovery from both short (sediment pulse) and long-term (damming) disturbance depends on a high level of ecosystem resilience [70]. In river systems, a key element of resilience is connectivity over multiple spatial domains—longitudinally along the river continuum, laterally across the floodplain, and vertically via groundwater and surface water exchange [71–73]. Connectivity supports high levels of habitat complexity, and generates multiple pathways by which energy moves between habitats and across trophic levels. With 83% of the watershed protected as wilderness inside of Olympic National Park, the Elwha River maintains a high level of floodplain connectivity. Now with the removal of the two dams, longitudinal connectivity is also in place.

The shifts we observed in food web structure during dam removal illustrate various mechanisms by which connectivity confers resilience. The increased importance of terrestrial food sources during dam removal demonstrates the strength of riparian-aquatic linkages [41,44,74]. The presence of many aquatic invertebrate taxa in fish diet despite their absence from our environmental sampling during dam removal suggest that these invertebrates persisted in areas we did not sample (such as sub-surface habitats or groundwater channels). And the rapid reappearance of many aquatic invertebrate taxa by the end of active dam removal reflects nearby source populations—both from floodplain habitats and less-impacted upstream reaches.

Other species in the greater Elwha food web are also responding to shifting resource availability. Bull trout in the Elwha have resumed long-distance migrations from the headwaters to the estuary, resulting in increased body mass at length [75,76]. The riparian songbird American dipper (*Cinclus mexicanus*) was significantly enriched in salmon-derived nutrients almost immediately following dam removal, which translated into increased reproductive success [77,78]. Wildlife and vegetation have also moved into the two former reservoir areas, with positive increases in both on the former lakebed surfaces [39].

As we move further along the response trajectory, we hypothesize that food web complexity will continue to increase as annual sediment loads approach natural background levels, and as

anadromous fish recolonize the watershed between and upstream of the former dams. Both processes are already well underway. As of 2018, hydrogeomorphic metrics were trending towards a quasi-equilibrium state [5], and adult coho, Chinook, steelhead, bull trout, and lamprey have been documented upstream of the former Glines Canyon Dam [69,76]. As these populations continue rebuilding, additional energy sources in the form of marine-derived nutrients will further shape food web response to dam removal. Continuing to examine how this energy moves throughout the food web will help us better predict outcomes of future dam removal and river restoration projects [79].

## Supporting information

**S1 Fig. Shade plot of estuary invertebrate density over time.** Taxa are shown for benthic grab (a,b) and fallout samples (c,d) that contributed most to differences between years in spring (a,c) and summer (b,d), based on the SIMPER routine in PRIMER ($\geq$ 2% dissimilarity). Mean numerical density is fourth root transformed.
(TIFF)

**S2 Fig. Shade plot of river invertebrate density over time.** Taxa are shown that contributed the most to differences between years in (a) spring and (b) summer river drift, based on the SIMPER routine in PRIMER ($\geq$ 2% dissimilarity). Mean numerical density by river section and year are square-root transformed.
(TIFF)

**S3 Fig. Shade plot of estuary fish diet abundance over time.** Taxa are shown that contributed the most to differences between years in (a) spring, and (b) summer estuary fish diet, based on the SIMPER routine in PRIMER ($\geq$ 2% dissimilarity). Mean joules by season and year are fourth root transformed.
(TIFF)

**S4 Fig. Shade plot of river fish diet abundance over time.** Taxa are shown that contributed the most to differences between years in a) spring, and b) summer estuary fish diet, based on the SIMPER routine in PRIMER ($\geq$ 2% dissimilarity). Mean joules by season and year are fourth root transformed.
(TIFF)

**S1 Table. Aquatic-origin invertebrates collected in environmental samples, by study section.**
(PDF)

**S2 Table. Terrestrial-origin invertebrates collected in environmental samples, by study section.**
(PDF)

**S3 Table. Body measurement to dry mass (DM) conversions for all taxa encountered in environmental and fish diet samples.**
(PDF)

**S4 Table. Energy density and percent dry mass (pDM) conversions for all taxa encountered in environmental and fish diet samples.**
(PDF)

## Acknowledgments

This project would not have been possible without field assistance from innumerable staff, interns, and volunteers from the Lower Elwha Klallam Tribe, NOAA's Northwest Fisheries Science Center, U.S. Geological Survey (USGS) Western Fisheries Research Center, U.S. Fish and Wildlife Service Western Washington Fish and Wildlife Conservation Office, Peninsula College School of Fisheries Technology, Washington Conservation Corps, and the Coastal Watershed Institute. We thank Ms. Josephine Pederson and Green Crow Timber for access to private lands. Scott Anderson (USGS) provided assistance with the stage-gage analysis of Fig 2. Taxonomic identification of invertebrates was provided by Arthur Frost, Aquatic Biology Associates, and Rhithron Associates. Invertebrate biomass conversion equations presented in Supplemental Materials grew out of a literature review first initiated by Dave Rundio at NOAA's Southwest Fisheries Science Center and continued by Dr. Sean Naman at the University of British Columbia. Sean Sullivan (Rhithron Associates) provided assistance developing biomass equations for taxa not found in the literature. We thank Josh Chamberlin, Dave Rundio, Steve Rubin, and Mark Sorel for their thoughtful reviews on earlier versions of this manuscript. Any use of trade, firm, or product names is for descriptive purposes only and does not imply endorsement by the U.S. Government. Data generated in this study and associated efforts monitoring environmental metrics before and during Elwha dam removal are available in Foley et al. [80] and at Morley et al. [81].

## Author Contributions

**Conceptualization:** Sarah A. Morley, Melissa M. Foley, Jeffrey J. Duda, Rebecca L. Paradis, Michael L. McHenry.

**Data curation:** Sarah A. Morley, Melissa M. Foley, Rebecca L. Paradis, Rachelle C. Johnson, Justin Stapleton.

**Formal analysis:** Sarah A. Morley, Melissa M. Foley, Jeffrey J. Duda, Rachelle C. Johnson.

**Funding acquisition:** Melissa M. Foley, Jeffrey J. Duda, Mathew M. Beirne, Rebecca L. Paradis, Michael L. McHenry, George R. Pess.

**Investigation:** Sarah A. Morley, Melissa M. Foley, Jeffrey J. Duda, Mathew M. Beirne, Rebecca L. Paradis, Mel Elofson, Earnest M. Sampson.

**Methodology:** Sarah A. Morley, Melissa M. Foley, Mathew M. Beirne, Rebecca L. Paradis, Justin Stapleton.

**Project administration:** Mathew M. Beirne, Michael L. McHenry, George R. Pess.

**Resources:** Rebecca L. Paradis, Michael L. McHenry, Mel Elofson, Earnest M. Sampson, George R. Pess.

**Software:** Randall E. McCoy.

**Supervision:** Sarah A. Morley, George R. Pess.

**Visualization:** Sarah A. Morley, Melissa M. Foley, Jeffrey J. Duda, Rachelle C. Johnson, Randall E. McCoy.

**Writing – original draft:** Sarah A. Morley, Melissa M. Foley, Jeffrey J. Duda, Mathew M. Beirne, Rachelle C. Johnson.

**Writing – review & editing:** Sarah A. Morley, Melissa M. Foley, Jeffrey J. Duda.

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
