## [Decision Letter · Decision Letter 0]

24 Jun 2020

PONE-D-20-06632

Shifting food web structure during dam removal—disturbance and recovery during a major restoration action

PLOS ONE

Dear Dr. Morley,

Thank you for submitting your manuscript to PLOS ONE. After careful consideration, we feel that it has merit but does not fully meet PLOS ONE’s publication criteria as it currently stands. Therefore, we invite you to submit a revised version of the manuscript that addresses the points raised during the review process.

I got the recommendations and comments from an expert reviewer on the field. The  reviewer agreed that the manuscript is technically sound and the data support the conclusions.However, lack of the explanation in Introduction and method sections were suggested, and I totally share their comments. Therefore, I can invite you to submit a revised version of the manuscript that addresses the points raised by the reviewer.

We look forward to receiving your revised manuscript.

Kind regards,

Hideyuki Doi

Academic Editor

PLOS ONE

Additional Editor Comments:

I got the recommendations and comments from an expert reviewer on the field. The reviewer agreed that the manuscript is technically sound and the data support the conclusions.However, lack of the explanation in Introduction and method sections were suggested, and I totally share their comments. Therefore, I can invite you to submit a revised version of the manuscript that addresses the points raised by the reviewer.

Journal Requirements:

2. In your Methods section, please provide additional location information of the collection sites, including geographic coordinates for the data set if available.

3. We noted in your submission details that a portion of your manuscript may have been presented or published elsewhere.

"Estuary benthic and shoreline invertebrate data was previously published as part of Foley et al. 2017 "Coastal habitat and biological community response to dam removal on the Elwha River". We have included it in this submission because (1) it informs the estuary fish diet data composition presented here for the first time, and (2) it allows for side-by-side comparison of estuary versus riverine invertebrate response to dam removal. Invertebrate data previously published in Foley et al. 2017 was also presented at a coarser taxonomic level (Order) than analyzed in this submission."

5. We note that Figure 1 in your submission contain map images which may be copyrighted. All PLOS content is published under the Creative Commons Attribution License (CC BY 4.0), which means that the manuscript, images, and Supporting Information files will be freely available online, and any third party is permitted to access, download, copy, distribute, and use these materials in any way, even commercially, with proper attribution. For these reasons, we cannot publish previously copyrighted maps or satellite images created using proprietary data, such as Google software (Google Maps, Street View, and Earth). For more information, see our copyright guidelines: http://journals.plos.org/plosone/s/licenses-and-copyright.

5.1.    You may seek permission from the original copyright holder of Figure 1 to publish the content specifically under the CC BY 4.0 license.

5.2.    If you are unable to obtain permission from the original copyright holder to publish these figures under the CC BY 4.0 license or if the copyright holder’s requirements are incompatible with the CC BY 4.0 license, please either i) remove the figure or ii) supply a replacement figure that complies with the CC BY 4.0 license. Please check copyright information on all replacement figures and update the figure caption with source information. If applicable, please specify in the figure caption text when a figure is similar but not identical to the original image and is therefore for illustrative purposes only.

Reviewers' comments:

Reviewer's Responses to Questions

**Comments to the Author**

1. Is the manuscript technically sound, and do the data support the conclusions?

Reviewer #1: Yes

2. Has the statistical analysis been performed appropriately and rigorously? 

Reviewer #1: I Don't Know

3. Have the authors made all data underlying the findings in their manuscript fully available?

Reviewer #1: No

4. Is the manuscript presented in an intelligible fashion and written in standard English?

Reviewer #1: No

5. Review Comments to the Author

Reviewer #1: General Comments

This manuscript contains a massive amount of information and provides a comprehensive picture of some of the impacts of sedimentation during dam-removal on invertebrate prey availability and salmonid diets. I do think it needs to be edited further. See detailed comments below for some examples.

Furthermore, given the amount of information presented, any streamlining would make the paper easier to read. This paper is unconventional in its length and breadth. This is not necessarily a bad thing. A more conventional treatment might be to split this into multiple papers and focus the story on a specific aspect of the larger story in each. I will not say that either approach is superior, and certainly there are advantages to both. This broad style of paper provides all the relevant information in one spot, and supports big picture thinking, which is important in ecology and arguably we need to do more of it. However, a shorter paper is faster and easier to process when considering one aspect of the bigger story. It may also be possible to have multiple shorter papers that build upon one another. It may be possible to remove electivity, CPUE, and condition index, and still tell much of the current story.

If the current breadth is kept, I think that the introduction needs to better set the stage for what will be in the paper. There is some detailed analysis in this paper about electivity and the like, which are never mentioned in the introduction. See comment below.

Lastly, it is impossible to tell if the data are available, because not enough detail is given to locate them. More information is needed on the location of the data beyond https://www.sciencebase.gov/catalog/ and https://www.data.gov/

Detailed Comments

Could background on Elwha Dam removal be condensed?

Line 131 replace changed with varied?

Because the paper covers so many topics (e.g., prey availability, diet, selectivity) It might be nice to set the stage in the introduction for why these matter and how they are all related. As is, intro is focused on background on dam removal and sediment, without much discussion of biology. Makes it a bit surprising when things like electivity, CPUE, and condition factor are brought in during the methods.

216 Were velocities measured at the beginning and end of the sample period?

254 perhaps change “sample metrics” to something more informative. You talk about statistical analysis in this section too.

261 “section 3.4” ?

268-277 Not a critique on scientific soundness, but this paper is already quite complex and large. Is adding CPUE and condition important for the overall story. Should it be mentioned in the introduction?

271. is CPUE in the river meaningful if no consideration for effort (in terms of time) other than the 90 minutes max effort. Maybe it is if it was rare to get the 10 fish within the 90 minutes.

282. does it make sense to include empty stomaches? Potentially conflating diet composition similarity, and “fulness” similarity. The fish with empty stomachs are eating something, just not when they were sampled. Therefore, I would probably just exclude the empty stomachs form the analysis.

278-284, assessing similarity of diet and ambient prey, and selectivity, was not mentioned in objectives.

285, index of relative Importance not Abundance

289 I believe this is different than the traditional IRI from Hart et al 2002, which is F(N+V), where F = frequency of occurrence percentage, N = numerical percentage, and V = volumetric percentage. But, I’m not sure what the “mean relative numeric abundance of order i” or “mean relative energetic abundance of order i” are. Maybe they are the same as percentages? If they are different, it would require some explanation for why the modification was made. Also, is %F frequency of occurrence percentage, or just frequency of occurrence, as written?

295 Hart et al 2002 sat that IRI should not be compared across groups, but rather ranks of different prey types should be compared. The method of standardizing IRI and then comparing may be valid, but there should be a citation to support it or some explanation for why it is valid. Since I can’t really tell what the calculated IRI is, I definitely can’t tell if this method of comparing across groups is valid.

296-298 No citation indicates that you developed this method of evaluating selectivity? It seems like it might be valid, but there are other more established electivity indices.

306 What was the critical value used for Tukeys HSD test?

313 How were age classes determined? sample sizes?

317 why are “section” and “year” capitalized

345 where do energy density values come from in table 2?

416. of course things “changed” in the reference section in some way or another. In some instances, it looks like there was more change in the control than in the sediment impacted sections. More detail would help. Is this referring to statistical significance?

427. Except summer ME increased.

496 consider changing “prey availability” to “energy availability”

545 Figure 11. Perhaps mention that a couple stress values are >.2

622 what do the different letters represent?

809 might these smolt data be a better indication of fish densities than your river % CPUE?

820 could turbidity reduce predation risk and therefore increase the duration or locations where salmonids can “safely” forage?

819 any “pink salmon year” effect?

829 5.3.1 sediment?

6. PLOS authors have the option to publish the peer review history of their article (what does this mean?). If published, this will include your full peer review and any attached files.

Reviewer #1: **Yes: **Mark Sorel

---

## [Author Response · Author response to Decision Letter 0]

20 Aug 2020

All reviewer and editor comments are addressed in the attached "Response to Reviewers" doc

---

## [Editor Report · Decision Letter 1]

2 Sep 2020

Shifting food web structure during dam removal—disturbance and recovery during a major restoration action

PONE-D-20-06632R1

Dear Dr. Morley,

We’re pleased to inform you that your manuscript has been judged scientifically suitable for publication and will be formally accepted for publication once it meets all outstanding technical requirements.

Kind regards,

Hideyuki Doi

Academic Editor

PLOS ONE

Additional Editor Comments (optional):

I carefully checked the revised manuscript as well as the response letter. I agree the revisions according to the reviewers’ comments and now can recommend to publish the paper in PLOS ONE.
---

## [Editor Report · Acceptance letter]

14 Sep 2020

PONE-D-20-06632R1 

Shifting food web structure during dam removal—disturbance and recovery during a major restoration action 

Dear Dr. Morley:

I'm pleased to inform you that your manuscript has been deemed suitable for publication in PLOS ONE. Congratulations! Your manuscript is now with our production department. 

Kind regards, 

on behalf of

Dr. Hideyuki Doi 

Academic Editor

PLOS ONE